# Chemically activating MoS$_2$ via spontaneous atomic palladium interfacial doping towards efficient hydrogen evolution

Zhaoyan Luo[1,2], Yixin Ouyang[3], Hao Zhang[4], Meiling Xiao[1], Junjie Ge[1], Zheng Jiang [4], Jinlan Wang[3,5], Daiming Tang[6], Xinzhong Cao[7], Changpeng Liu[1] & Wei Xing[1]

Lacking strategies to simultaneously address the intrinsic activity, site density, electrical transport, and stability problems of chalcogels is restricting their application in catalytic hydrogen production. Herein, we resolve these challenges concurrently through chemically activating the molybdenum disulfide (MoS$_2$) surface basal plane by doping with a low content of atomic palladium using a spontaneous interfacial redox technique. Palladium substitution occurs at the molybdenum site, simultaneously introducing sulfur vacancy and converting the 2H into the stabilized 1T structure. Theoretical calculations demonstrate the sulfur atoms next to the palladium sites exhibit low hydrogen adsorption energy at –0.02 eV. The final MoS$_2$ doped with only 1wt% of palladium demonstrates exchange current density of 805 μA cm$^{-2}$ and 78 mV overpotential at 10 mA cm$^{-2}$, accompanied by a good stability. The combined advantages of our surface activating technique open the possibility of manipulating the catalytic performance of MoS$_2$ to rival platinum.

[1] State Key Laboratory of Electroanalytical Chemistry, Jilin Province Key Laboratory of Low Carbon Chemical Power, Changchun Institute of Applied Chemistry, Chinese Academy of Sciences, 130022 Changchun, China. [2] University of Science and Technology of China, 230026 Hefei, Anhui, China. [3] School of Physics, Southeast University, 211189 Nanjing, China. [4] Shanghai Synchrotron Radiation Facility, Shanghai Institute of Applied Physics, Chinese Academy of Sciences, 201204 Shanghai, China. [5] Synergetic Innovation Center for Quantum Effects and Applications (SICQEA), Hunan Normal University, 410081 Changsha, China. [6] National Institute for Materials Science, Namiki 1-1, Tsukuba, Ibaraki 305-0044, Japan. [7] Institute of High Energy Physics, Chinese Academy of Sciences, 100049 Beijing, China. These authors contributed equally: Zhaoyan Luo, Yixin Ouyang. Correspondence and requests for materials should be addressed to J.G. (email: gejj@ciac.ac.cn) or to Z.J. (email: jiangzheng@sinap.ac.cn) or to J.W. (email: jlwang@seu.edu.cn) or to W.X. (email: xingwei@ciac.ac.cn)

The lack of a cost-effective replacement for Pt has plagued the scale-up of hydrogen electrochemical production ($2H^+ + 2e^- \rightarrow H_2$) for decades; the alternative catalytic materials are fundamentally limited by either a low catalytic efficiency or a short lifetime[1, 2]. Lamellar $MoS_2$ has been regarded highly promising towards hydrogen evolution reaction (HER) since the activity of its metallic edges ($\Delta G_H = 0.06$ eV) was theoretically predicted by Norskøv and co-workers[3], and was later on proved experimentally by Jaramillo and co-workers[4, 5]. The current guiding principles for advancing the $MoS_2$ catalytic efficiency are as follows: First, increase the atomically under-coordinated active sites density in the trigonal prismatic phase (2H) $MoS_2$, either through the preferentially exposing edge sites or through creating in-plane sulfur vacancies (SVs)[6–8]. However, unleashing the intrinsically high activity is retarded by the semiconductive feature of 2H–$MoS_2$, where the charge transfer efficiency is limited by a deficiency of electrons at the reaction interface[9, 10]. Second, drive the 2H phase $MoS_2$ into the conductive and therefore more catalytically active 1T phase[11, 12]. The basal-plane S atoms are regarded as active sites in 1T–$MoS_2$; however, these S sites suffer from less favorable hydrogen adsorption features ($\Delta G_H = 0.17$ eV) despite the greatly increased site density[13]. Beyond the above mentioned problems of 2H–$MoS_2$ and 1T–$MoS_2$, one major issue that both these materials encounter is their reduced stability because defective 2H–$MoS_2$ suffers from a high sulfur leaching rate[14] and 1T–$MoS_2$ is intrinsically metastable[15]. Apparently, $MoS_2$ only become truly applicable towards the HER when the electronic conductivity, site density, intrinsic activity, and stability issues are simultaneously solved.

Here, we report a highly active and long-life $MoS_2$-based HER catalyst, which is achieved by chemically activating its surface basal plane. We accomplish this by devising a thermodynamically spontaneous interfacial $MoS_2$/Pd (II) redox reaction. Pd was atomically doped into the original Mo sites, causing the generation of SVs, the conversion to the stabilized 1T phase, the stabilization of defective sites, and the intrinsic activation of the 1T basal plane. The final Pd–$MoS_2$ exhibits the highest HER performance ever achieved on heteroatom-doped $MoS_2$-based materials in an acidic solution, along with good cycling stability and an exceptional anti-leaching feature that exceed those of undoped $MoS_2$.

## Results

**Design of Pd–$MoS_2$.** We began by recognizing the redox characteristics of the chemically synthesized $MoS_2$. Unlike the standard $MoS_2$ samples, chemically synthesized $MoS_2$ generally contains a certain concentration of defects, thus resulting a final Mo to S stoichiometry deviated from the theoretical ratio of 1:2. According to the principles of the conservation of charge in defect chemistry, a mixed valence can thus be created, thereby endowing $MoS_2$ with redox power. We synthesized $MoS_2$ through a typical wet chemical method (homemade, $MoS_2$-HM, further denoted as $MoS_2$ in the following, see the Methods section for the details). The sample was first characterized by inductively coupled plasma mass spectrometry (ICP-MS), which shows a Mo to S stoichiometry of 1:1.87 (Supplementary Table 1), corresponding to a Mo average valence state of 3.74. The X-ray absorption near-edge structure (XANES) test was further carried out, with the standard 2H–$MoS_2$ used as a reference sample. The Mo $L_3$-edge XANES results (Fig. 1a) demonstrate a decrease in the white line resonance strength in comparison to the standard 2H $MoS_2$ sample, confirming a reduction in the unoccupied density of state (DOS) of Mo 4d and an average valance state lower than IV.[16] The high-resolution Mo 3d X-ray photoelectron spectroscopy (XPS) measurement directly evidence the concurrent presence of Mo (III)

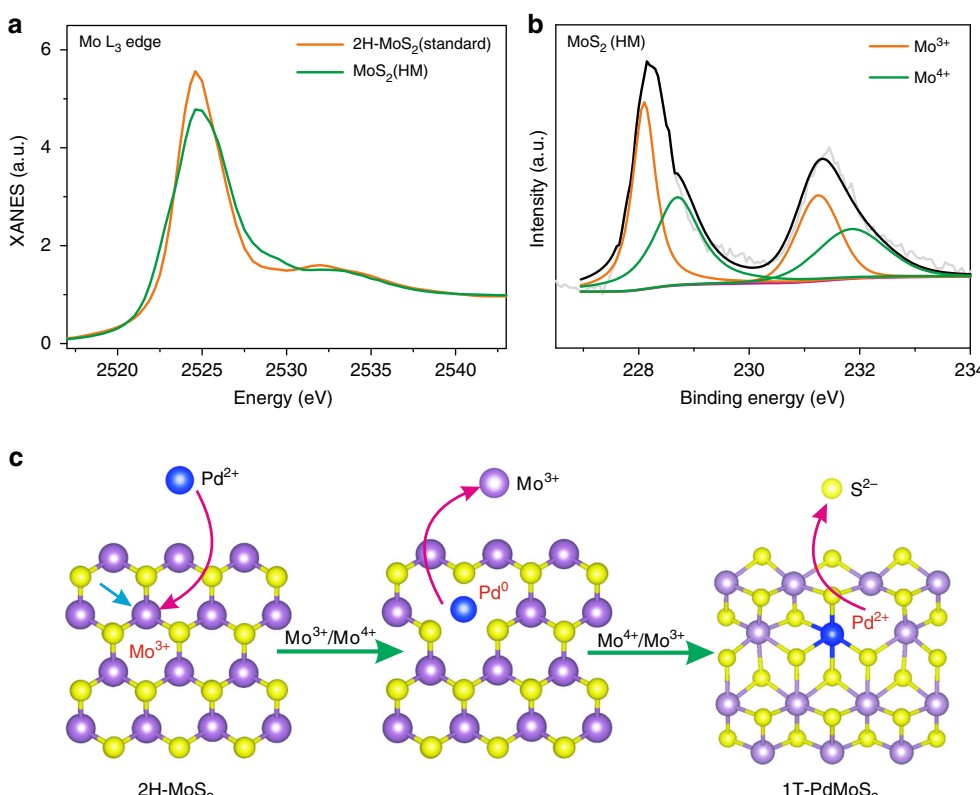

**Fig. 1** Design of a Pd–$MoS_2$ material based on recognizing the redox characteristics of $MoS_2$. **a** Mo $L_3$-edge XANES spectra of the homemade $MoS_2$. **b** High-resolution XPS results (Mo 3d region) of the homemade $MoS_2$. **c** Schematic illustration of the spontaneous $MoS_2$/Pd (II) redox reaction

and Mo (IV) (Fig. 1b) with deconvoluted doublets unambiguously assigned to Mo (III) ($3d_{3/2}$ at 231.5 eV and $3d_{5/2}$ at 228 eV) and Mo (IV) ($3d_{3/2}$ at 231.9 eV and $3d_{5/2}$ at 228.7 eV), respectively.

$$4Mo^{3+} \rightarrow 3Mo^{4+} + 3e(\text{left}) + V'''_{Mo}(\text{one Mo left}) \quad (1)$$

$$2Mo^{4+} + S^{2-} + 2e \rightarrow 2Mo^{3+} + V^{..}_{S}(\text{one S left}) \quad (2)$$

The Mo (III)/Mo (IV) couple represents a standard electrode potential at −0.04 V. If properly paired with another redox couple, the redox reaction is expected to occur at the MoS₂/liquid or gas interface. It is noted that structural vacancies can be created on the MoS₂ surface during Mo redox process, according to the principles of defect chemistry, i.e., the conservation of mass and charge, and the host structure. As shown in Eqs. 1 and 2, while Mo (III) oxidation creates Mo vacancies, Mo (IV) reduction leads to the formation of SVs. Meanwhile, inter-valence charge transfer between two ions[17] and abundant SVs[18] can also induce the phase transformation into the 1T structure. After carefully screening the transition elements, we deliberately chose Pd (II) to accomplish the interfacial reaction with MoS₂, as illustrated in Fig. 1c. A two-step thermodynamically spontaneous reaction is expected: First, the redox process in Eqn. 3 is a thermodynamically spontaneous reaction ($E^\theta = 1.031$ V, $\Delta rG^\theta = -198.98$ kJ, Supplementary Note 1), which leads to the reduction of Pd and oxidation of Mo, therefore, creating Mo vacancies due to the principles of conservation of charge. Afterwards, however, metallic Pd is thermodynamically favorable towards anchoring to the energetic Mo vacancies and spontaneously forming the more stable Pd–S bond ($Ksp=2.03 \times 10^{-58}$, Supplementary Table 2), as shown in Eq. 4, through its incorporation into the MoS₂ backbone ($E^\theta = 0.75$ V, $\Delta rG^\theta = -144.75$ kJ, Supplementary Note 1). By injecting electrons into the MoS₂ substrates, Mo (IV) is reduced back to Mo (III) and causes the leaching of S into the solution (law of conservation of charge) and the formation of S vacancies (see Eqs. 1–2). Thus, the MoS₂ basal plane is driven to spontaneously

incorporate Pd–S covalent bonds and form abundant SVs, presumably accompanied with phase conversion to form 1T-Pd–MoS₂.

$$2Mo^{3+} + Pd^{2+} \rightarrow 2Mo^{4+} + Pd$$
$$(E^\theta = 1.031 \text{ V}, \Delta rG^\theta = -198.98 \text{ kJ}) \quad (3)$$

$$Mo^{4+} + Pd + S^{2-} \rightarrow Mo^{3+} + PdS$$
$$(E^\theta = 0.75 \text{ V}, \Delta rG^\theta = -144.75 \text{ kJ}) \quad (4)$$

**Structure alteration due to Pd atomic doping.** The redox power of MoS₂ was first confirmed by a series of designed experiments between the pristine MoS₂ sample and the Pd (II), Pt (IV), and Au (III) complex solutions, where the detailed results and explanations are shown in Supporting Information (Supplementary Note 2, Supplementary Figs. 1-2, and Supplementary Tables 3-4). For the final Pd–MoS₂ samples, Pd with varied contents (1–15% Pd–MoS₂) were expectedly immobilized as Pd (II) in MoS₂, confirmed by the presence of the binding energy peaks at 336.7 eV ($3d_{5/2}$) and 342 eV ($3d_{3/2}$) (Fig. 2a). The Pd introduction does not induce observable morphological changes (Supplementary Figs. 3-4) to the MoS₂ nanosheets, and is (Supplementary Fig. 5) found homogeneously distributed by high-angle annular dark-field scanning transmission electron microscopy (HAADF-STEM) and elemental mapping, with no Pd-based crystalline phases observed (Fig. 2b). The results from in situ heat-treatment TEM coupled with electron energy loss spectroscopy (EELS, Supplementary Fig. 6) and X-ray diffraction (XRD, Supplementary Fig. 7) tests suggest that Pd was firmly integrated into the MoS₂ backbone without phase segregation even at 600 °C.

To elucidate the Pd local bonding environment and the occupation sites in MoS₂, extended X-ray absorption fine structure (EXAFS) and sub-angstrom resolution aberration-corrected HAADF-STEM were carried out. The Fourier

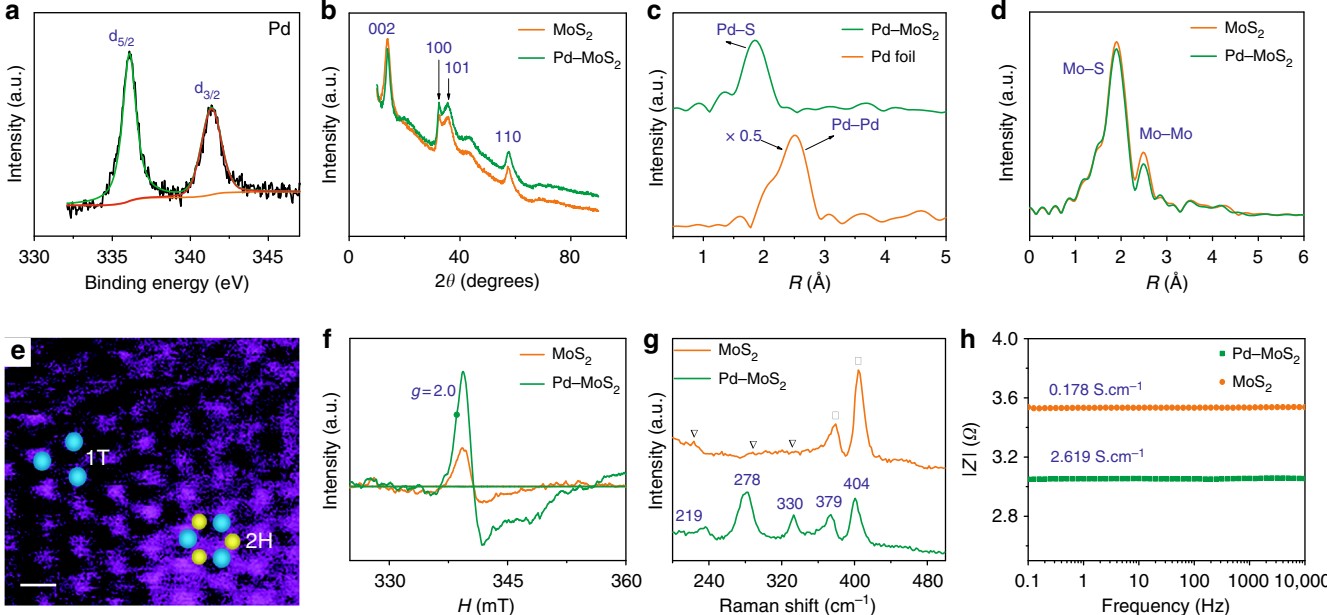

**Fig. 2** Structural characterization of 1%Pd–MoS₂ and MoS₂. **a** High-resolution XPS results (Pd 3d region) of the 1%Pd–MoS₂. **b** XRD patterns. **c** Fourier transform of the k³-weighted Pd K-edge of the EXAFS spectra. **d** Fourier transform of the k³-weighted Mo K-edge of the EXAFS spectra. **e** Dark-field scanning transmission electron microscopy image of the 1%Pd–MoS₂. Blue and yellow balls indicate Mo and S atoms, respectively. Scale bar: 1 nm. **f** ESR spectra. **g** Raman spectra of 1%Pd–MoS₂ and MoS₂. **h** Bode spectra obtained through electrochemical impedance spectroscopy with a frequency range from 0.1 Hz to 10 kHz and amplitude of 5.0 mV

transform of the $k^3$-weighted Pd K-edge EXAFS spectra (Fig. 2c, R-space) for Pd–MoS$_2$ shows the disappearance of the first-shell Pd–Pd scattering peak at 2.51 Å in comparison with Pd foils, indicating that the Pd species are formed neither as metallic Pd nanoparticles nor as Pd clusters. A prominent peak centered at a much lower R position is observed at 1.84 Å instead, corroborating the dominance of the Pd–S scattering contribution. We further fitted the main Fourier transform peaks from 1.1 to 2.5 Å in the R-space to quantitatively extract the Pd coordination parameters (Supplementary Fig. 8 and Supplementary Table 5). The best fit indicated a Pd–S bond distance of 2.33 Å and a Pd to S coordination number of 4.33, which is smaller than the nominal Mo–S coordination[19]. The Mo K-edge EXAFS spectra (Fig. 2d) reveal a decrease in the Mo–S and Mo–Mo peak intensities after Pd doping. While the former can be attributed to newly generated SVs, the latter may originate from the rearrangement of both atoms due to Pd fixation and the formation of SVs. The absence of first-shell Pd–Mo scattering in both Fig. 2c, d excludes the possibility that Pd was directly bound with Mo, thereby confirming that Pd does not occupy the S sites in MoS$_2$. The sub-angstrom resolution aberration-corrected HAADF-STEM (Supplementary Fig. 9) shows an ordered MoS$_2$ crystalline structure after Pd fixation, which is evident of the atomic dispersion of Pd. Thus, judging from the successful immobilization of Pd into the MoS$_2$ frame, the only remaining possibility is that Pd replaces the Mo site, as we expected.

We next examined the influence of Pd atomic doping on the formation of S vacancies by comparing the 1%Pd-MoS$_2$ with pristine MoS$_2$. Electron spin resonance (ESR Fig. 2f) is utilized to detect the paramagnetic signals, and the ~337 mT ($g = 2$) signal intensity reflects the concentration of unsaturated sites with unpaired electrons. The pristine MoS$_2$ demonstrates a relative intensity at $0.52 \times 10^3$ a.u. mg$^{-1}$ sites, corresponding to the edges, crystallite interfaces, and in-plane SVs[20]. Remarkably, the 1% Pd–MoS$_2$ exhibits a signal intensity ($1.55 \times 10^3$ a.u. mg$^{-1}$) approximately 3 times that of the MoS$_2$, corroborating the formation of abundant SVs, in accordance with Eqn. 2 and the EXAFS results (with an in-plane Pd–S bond coordination number of 4.3 and a decrease in the scattering strength for the Mo–S path). We further employed positron annihilation lifetime spectroscopy (PALS, Supplementary Fig. 10 and Supplementary Table 6) to understand the form and content of the defects. $\tau_1$ is assigned to the lattice defects, such as structural vacancies ($\tau_v$) or dislocation vacancies ($\tau_d$), while $\tau_2$ is caused by clusters of vacancies. Notably, both $\tau_1$ and $\tau_2$ increased (Table 1) after 1% Pd doping, i.e., from $183.6 \pm 5.3$ and $355.5 \pm 5.9$ ps to $206.2 \pm 4.7$ and $384.6 \pm 8.1$ ps, respectively, which corresponded to an increase in the defect dimension. At the same time, the $\tau_1$ and $\tau_2$ relative intensities, denoted as $I_1$ and $I_2$, are also tuned, showing an increase in $I_1$ from 49.1% to 57.5% and a decrease in $I_2$ from 49.2% to 40.8%, which suggested that more single lattice SVs than vacancy clusters are introduced. ICP-OES results show that the final 1%Pd–MoS$_2$ represents a stoichiometry of Pd$_{0.02}$MoS$_{1.82}$, with an S ratio significantly lower than that in the pristine MoS$_{1.87}$ sample. According to the XPS surface analysis, the total amount of surface SVs created due to Pd doping is estimated to be 16.7% (Supplementary Table 7).

The impact of Pd on the phase conversion was further investigated. Careful examination of the (Fig. 2e, Supplementary Fig. 11 and Supplementary Note 3) reveals structurally distinct domains in the 1%Pd–MoS$_2$ sample. A trigonal lattice structure corresponding to the 1T–MoS$_2$ and the common honeycomb lattice area of the 2H–MoS$_2$[21, 22] are both clearly visible, as indicated by the circles in Fig. 2e. However, the predominant fraction of 1T–MoS$_2$ demonstrates the heterogeneous structure (2H + 1T) of the Pd–MoS$_2$, and Raman spectroscopy provides direct evidence of the 2H to 1T phase conversion. Pristine MoS$_2$ (Fig. 2g) exhibits two peaks at 378 cm$^{-1}$ and 404 cm$^{-1}$, attributable to the 2H phase vibrational configurations of the in-plane Mo–S phonon mode ($E_{2g}$) and the out-of-plane Mo-S mode ($A_{1g}$), respectively[23–25]. In contrast, 1%Pd–MoS$_2$ exhibits new prominent peaks at 146, 278, and 332 cm$^{-1}$, which were obviously associated with the 1T-MoS$_2$ phonon modes, along with distinctly weakened 2H–MoS$_2$ signals ($E_{2g}$ and $A_{1g}$). As a result of the phase conversion, the electronic conductivity (Fig. 2h, see the Supplementary Methods for details) is more than one order of magnitude greater for 1%Pd–MoS$_2$ (2.619 S cm$^{-1}$ versus 0.178 S cm$^{-1}$ for MoS$_2$) and approached that of 1T–MoS$_2$ (10–100 S cm$^{-1}$).[26]

**The HER catalytic behavior.** Next, we examined the HER catalytic behavior of Pd–MoS$_2$ (1–15% doping), MoS$_2$, Pd/C, and commercial Pt/C catalysts with the representative linear sweep voltammograms (LSVs) summarized in Fig. 3a and Supplementary Fig. 12a-b. First, the pristine MoS$_2$ exhibited an overpotential of 10 mA cm$^{-2}$ ($\eta$@10 mA cm$^{-2}$) at 328 mV, being consistent with those reported for 2H–MoS$_2$ in the literature[27, 28]. Second, Pd doping leads to a breakthrough in the catalytic performance towards the HER, far exceeds that of the metallic Pd catalysts. The 1%Pd–MoS$_2$ exhibited a current density of 10 mA cm$^{-2}$ at an overpotential of only 89 mV. This result corresponds to the highest performance ever reported for heteroatom-doped MoS$_2$-based catalysts in acidic media[28–32], and 1%Pd–MoS$_2$ is the best among previously reported phase-pure MoS$_2$ based materials in the literature (see Supplementary Table 8 for details). Third, supporting Pd-MoS$_2$ on carbon paper ($\eta$@10 mA cm$^{-2}$ = 78 mV) further boost the activity to approach that demonstrated for Pt/C catalysts. Fourth, while increasing Pd from 1–10% (10% Pd–MoS$_2$, $\eta$@10 mA cm$^{-2}$ = 72 mV) results in increased activity towards the HER, further increasing the Pd doping content results (15%) in a decay in performance.

The corresponding Tafel plots (Fig. 3b and Supplementary Fig. 12c) show that Pd atomic doping decreases the Tafel slope from 157 to 62–80 mV dec$^{-1}$ (Pd-MoS$_2$, 1–10%), demonstrating the transition of the rate determining step (RDS) away from the Volmer discharge reaction ($H_3O^+ + e^- \rightarrow H_{ads} + H_2O$). This transition is expected since Pd doping results in a phase conversion to 1T–Pd–MoS$_2$, thus making access to the electrons and the formation of H$_{ads}$ easier at the interface. The exchange current densities ($j_0$) were further calculated to demonstrate the inherent HER activity, as shown in Fig. 3c. Pd doping is extremely effective for boosting the $j_0$ because 1%Pd–MoS$_2$ (805 μA cm$^{-2}$,

**Table 1 Summary of the electrochemical and structural properties of 1%Pd–MoS$_2$ and MoS$_2$ catalysts**

| Catalyst | $\eta$ (mV vs RHE) for | $j_{0, geometrical}$ | ESR intensity of | Position lifetime parameters | | | |
|---|---|---|---|---|---|---|---|
| | $j = -10$ mA cm$^{-2}$ | (μA cm$^{-2}$) | S ($\times 10^3$ a.u. mg$^{-1}$) | $\tau_1$ (ps) | $I_1$ (%) | $\tau_2$ (ps) | $I_2$ (%) |
| 1%Pd-MoS$_2$ | 89 | 805 | 1.55 | $206.2 \pm 4.7$ | $57.5 \pm 2.1$ | $355.5 \pm 5.9$ | $40.8 \pm 2.1$ |
| MoS$_2$ | 328 | 37.25 | 0.52 | $183.6 \pm 5.3$ | $49.1 \pm 1.1$ | $384.6 \pm 8.1$ | $49.2 \pm 1.9$ |

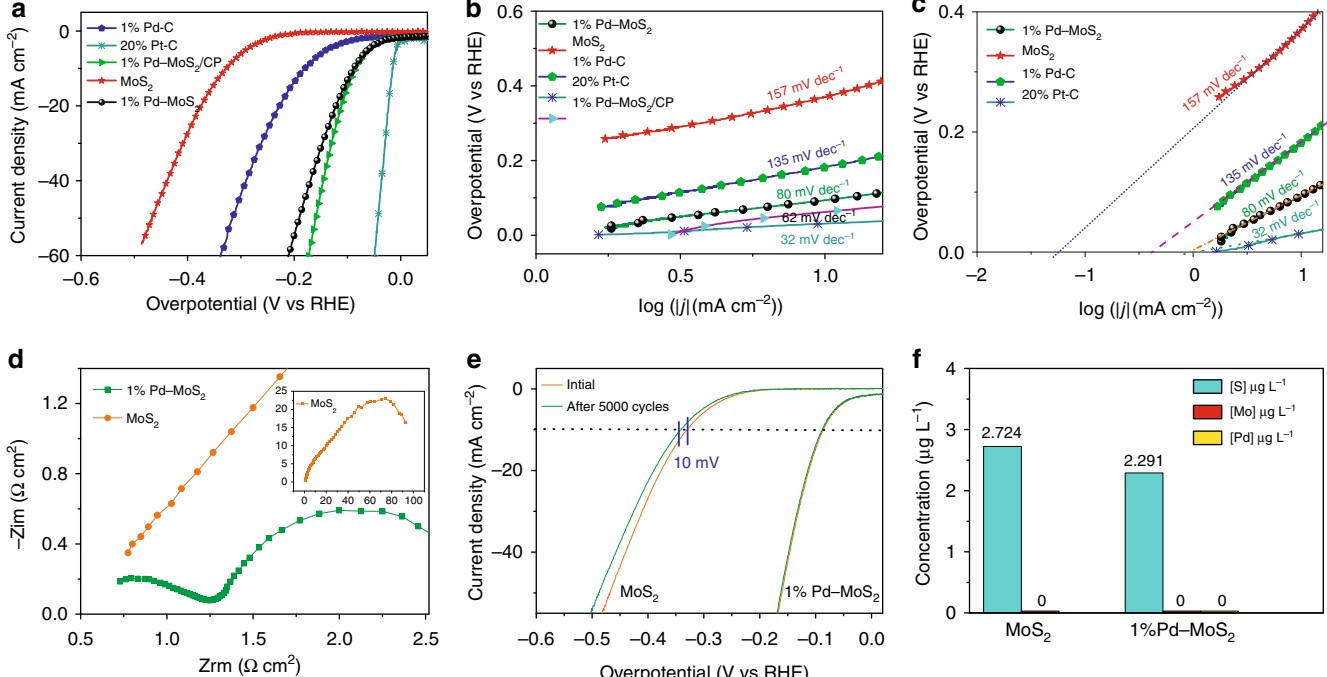

**Fig. 3** Superior activity and stability of 1%Pd–MoS$_2$. **a** LSV polarization curves of MoS$_2$, 1%Pd–MoS$_2$, 1%Pd–MoS$_2$/CP, 1%Pd–C, and 20%Pt–C (with iR correction). **b** Tafel plots derived from the results given in Fig. 3a. **c** Exchange current density for the MoS$_2$, 1%Pd–MoS$_2$, 1%Pd–C and 20%Pt–C samples, derived from the Tafel plots, as indicated by the dashed lines. **d** EIS comparison for the MoS$_2$ and 1%Pd–MoS$_2$ in terms of Nyquist plots; inset, the full-spectrum for the MoS$_2$. **e** Stability measurements for MoS$_2$ and 1%Pd–MoS$_2$ using accelerated degradation tests (5000 cycles, 100 mV s$^{-1}$); polarization curves are shown without iR correction. **f** ICP-OES results of dissolved S, Mo, and Pd ions in the electrolyte after the stability tests shown in Fig. 3e

the best of all reported MoS$_2$ based materials) is significantly better than MoS$_2$ (37.25 μA cm$^{-2}$) and almost reached that of 20% Pt/C (918 μA cm$^{-2}$) at the same catalyst loading[7,27,33]. The electrochemical impedance spectroscopy (EIS) results explain the exceptional HER behavior of 1%Pd–MoS$_2$, where the Nyquist plots (Fig. 3d) show a large reduction in the charge transfer resistance ($R_{ct}$) from 92.89 Ω cm$^2$ for MoS$_2$ to 1.50 Ω cm$^2$ (Supplementary Fig. 13 and Supplementary Table 9) for 1% Pd–MoS$_2$.

In addition to having a high HER activity, 1%Pd–MoS$_2$ is a stable and long-life catalyst. We combined the electrochemical measurements and ICP-OES test to verify the material durability during operation. The chronoamperometry test results (Supplementary Fig. 14) manifest that the 1%Pd–MoS$_2$ exhibits an outstanding long-term operational stability beyond 100 h with an observed potential increase of only 14 mV. The ultrahigh stability of 1%Pd–MoS$_2$ was also proved by long-term cyclic voltammetry tests, where no obvious potential decay was observed (Fig. 3e) after 5000 cycles. In contrast, the MoS$_2$ electrode shows that $\eta$@10 mA cm$^{-2}$ increased by 10 mV after 5000 cycle tests. The ex situ ICP-OES results present a reduced sulfur concentration (2.724 versus 2.291 ppm) in the testing electrolyte (Fig. 3f), which indicated that the final defective 1%Pd–MoS$_2$ surface is even more stable than that of the pristine MoS$_2$. Moreover, we further examined the Pd–MoS$_2$ catalyst using XPS characterization after the above electrolysis test (Supplementary Fig. 15). Neither the content nor the state of Pd was altered, suggesting that Pd is firmly integrated into the MoS$_2$ backbone and highly stable under electrolytic conditions. This result is contradicted with those reported in the literature because highly active and defective catalysts are always accompanied by poor stability. Thus, by tailoring the chemical bond characteristics, we designed a MoS$_2$ material which simultaneously possesses good stability and activity.

**Density functional theory calculations**. Density functional theory (DFT) calculations were carried out to obtain atomic-scale insight into the doping effect of Pd. First, the energies of the Pd atoms on varied sites were calculated, and it is found that Pd exhibits a strong tendency to replace Mo with an exothermic energy of −2.22 eV compared to replacing S (−0.17 eV) adsorbed on the Mo atop site (1.75 eV) and the hollow site (2.35 eV) (see Supplementary Note 4 and Supplementary Fig. 16), thus supporting our expectation of the thermodynamically driven formation of the Pd–Mo–S$_x$ compound. Second, we calculated the energy for the formation of SVs in MoS$_2$ and Pd–MoS$_2$ (Fig. 4b and Supplementary Fig. 17), and the energy for SVs formation decreased by ~1–2 eV due to the Pd doping. Thus, we can use the spontaneous Pd doping strategy to create a large number of SVs on the MoS$_2$. Third, we explored the influence of Pd doping and the SV concentration on the total phase energy of both the 1T and 2H MoS$_2$. In Fig. 4a-b, we found that 1T–MoS$_2$ becomes more stable than 2H–MoS$_2$ (see Supplementary Note 5 for the detailed results) with the presence of the Pd and SVs at certain concentrations. Specifically, the 1%Pd-MoS$_2$sample, as confirmed by the XPS (Supplementary Table 7) to possess surface concentrations of Pd at 3.47% and newly generated SVs at 16.7%, corresponding to the higher stability of the 1T versus 2H phase shown in Fig. 4b. Finally, we investigated the effect of Pd doping on the HER activity of the MoS$_2$ basal plane. The hydrogen adsorption free energy ($\Delta G_H$) was used to evaluate the hydrogen evolution activity. The Pd sites themselves were calculated to be inactive as H does not form a very stable adsorption structure on Pd atop site (Supplementary Fig. 18). The $\Delta G_H$ of the SVs in the 1T MoS$_2$ is 0.09 eV, suggesting more favorable HER catalytic behavior than that of 1T basal-plane ($\Delta G_H$ = 0.17 eV). Therefore, the increases in SVs concentration unambiguously contribute to the increased catalytic behavior, and we regard this as an increase in the site density. More excitingly, the $\Delta G_H$ of the S atop site

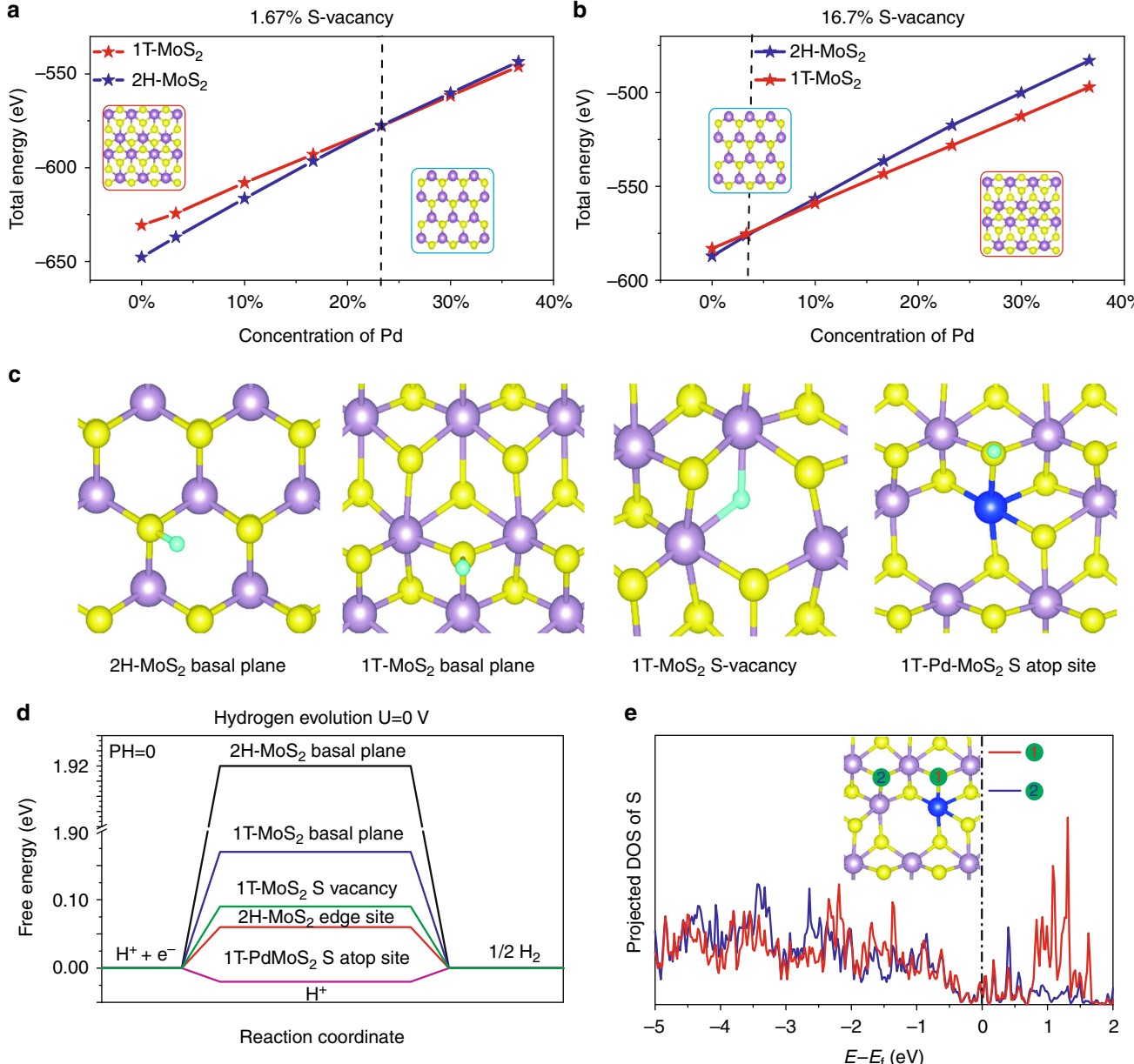

**Fig. 4** DFT calculation for the energetics of Pd doping to promote the HER activity of MoS$_2$ variation in the energy of 2H–MoS$_2$ and 1T–MoS$_2$ as the concentration of Pd changes at SV concentrations of **a** 1.67% and **b** 16.7%. **c** Adsorption positions for a single H atom absorbing on pure MoS$_2$ and Pd–MoS$_2$. **d** Free energy versus the reaction coordinates of different active sites. **e** Projected p-orbital density of states of S. Site 1 indicates S atop site adjacent to the Pd atoms in the 1T MoS$_2$; site 2 indicates the in-plane S of the 1T basal plane

adjacent to Pd (Pd–S*–Mo) in the 1T–Pd–MoS$_2$ exhibits an almost thermoneutral value of −0.02 eV, which was far better than the reported most favorable edge sites (Fig. 4c, d and Supplementary Fig. 19)[4, 34]. In Fig. 4e, the difference in the S atom electronic structures of site 1 (Pd adjacent S site) and site 2 (regular in-plane S site) were elucidated by the projected p-orbital DOSs of S. Because the ionization energy of Mo is less than that of Pd, site 2 receives more electrons from the adjacent three Mo atoms with more p-states filled, which leads to a weaker interaction with hydrogen[35–37] and hence a higher free energy for hydrogen adsorption. Considering that the HER performance is enormously increased and is far better than those reported for 1T–MoS$_2$ with abundant edge and vacancy sites[38], we believe that the significant increase in the catalytic activity is due to a combination of the 2H to 1T phase transition, the increase in SVs sites, and the introduction of new Pd–S* sites.

## Discussion

By combining the experimental and theoretical calculation results, the high catalytic efficiency and the long life-time of the Pd–MoS$_2$ is confirmed, and the reason for the good HER catalytic feature can be rationally explained. We ascribe the fast Faradaic process to the following reasons: first, the facilitated electrical transport due to the phase conversion in Pd–MoS$_2$; nevertheless, this cannot be the sole reason since the best activity of pure 1T–MoS$_2$ reported in the literature was only $\eta$@10 mA cm$^{-2}$ = 187 mV[12]. Second, the creation of abundant SVs, which are catalytically more active than the in-plane S sites in 1T–MoS$_2$. It is noted that even in the case of 1T–MoS$_2$ with abundant S vacancies and edges (the degree of defects exceeds 83.7%), the lowest overpotential ever reported was 153 mV[38]. Moreover, the 1%Pd–MoS$_2$ exhibits high turnover frequency (TOF) per active site (Supplementary Note 6, Supplementary Table 10), which

surpasses most state-of-the-art $MoS_2$ reported (Supplementary Table 11). Therefore, the exceedingly high HER performance of our catalysts is believed to only partially originate from the SVs in the 1T-$MoS_2$. Third, the generation of intrinsically more active Pd–S* sites with an almost thermoneutral $\Delta G_{H}$, (see Supplementary Table 12 and Supplementary Note 7 for details).

Though the positive effect of Pd doping on $MoS_2$ activation is confirmed, a Pd doping content that was too high (>15%) resulted in the decay in performance, which can be explained as follows: in the Pd–$MoS_2$ catalyst, Pd atoms are not the active sites, and instead, they function by activating the $MoS_2$ basal plane through the introduction of SVs and by activating the adjacent S atoms (Pd–S*–Mo). However, a Pd content that is too high will result in a decrease in the site density of Pd–S*–Mo, and a S-vacancy concentration that is too high results in deviation of intrinsic activity from the optimal value[39, 40], both of which hinder the final HER performance.

The high in situ operating stability can be ascribed to two reasons: first, although 1T–$MoS_2$ is believed to be a metastable phase by itself, the incorporation of Pd makes the 1T–$MoS_2$ more stable than 2H–$MoS_2$ due to the simultaneous presence of Pd and SVs; second, the high Pd–S bond stability can provide overall stability. The ultralow solubility product of PdS renders the Pd–S bond 20 magnitudes more stable than other transition metal sulfides (Supplementary Table 2). Thus, the exposed under-coordinated in-plane sites are prevented from further leaching, thereby leading to a stabilized defective surface.

In summary, we chemically activated $MoS_2$ by thermodynamically spontaneous Pd atomic doping. This is the first report to unveil the $MoS_2$ redox feature and use it to induce interfacial heteroatom doping. We found that Pd replaces Mo at the surface, bonds covalently with S atoms at a coordination number of 4.3, causes SVs formation and phase conversion, and strongly activates the neighboring S sites for HER. The Pd–$MoS_2$ catalyst exhibits the highest activity towards the HER among phase-pure $MoS_2$ based materials in acid media. The overpotential for 1%Pd–$MoS_2$ at 10 mA cm$^{-2}$ is only 78 mV cm$^{-2}$. More attractively, the more defective Pd–$MoS_2$ demonstrates better matrix stability than the pristine $MoS_2$. Therefore, the catalytic efficiency and stability problems for $MoS_2$ are addressed at the same time, leading to a promising future in replacing Pt-based electrocatalysts for the HER.

## Methods

**Materials**. The ammonium molybdate tetrahydrate (($NH_4$)$_6$$Mo_7O_{24}$•$4H_2O$), thiourea ($CH_4N_2S$), palladium acetate (Pd(OAc)$_2$), chloroplatinic acid ($H_2PtCl_6$•$6H_2O$), Chlorchloric acid ($H_2AuCl_4$), and 5wt% Nafionionomer was purchased from Aldrich. Commercial 20wt% Pt/C (HiSPEC™ 3000, denoted as Pt/C-JM) was purchased from Johnson Matthey Company. Vulcan carbon black (XC-72) was purchased from Cabot Co. All of the chemicals were used directly without further treatment or purification. Highly purified argon (≥99.99%) was from Changchun Juyang Co Ltd. Ultrapure water with resistivity higher than 18 MΩ cm$^{-1}$ was used in all the experiments.

**Materials synthesis**. The HM–$MoS_2$ material was synthesized through a solvothermal method. Firstly, 0.5213 g of ammonium molybdate tetrahydrate (($NH_4$)$_6$$Mo_7O_{24}$.$4H_2O$) and 1.035 g of thiourea ($CH_4N_2S$) were dissolved in 30 ml of water in a beaker and then sonicated for 1 h. The resulting homogenous solution was transferred into a 50 ml Teflon-lined stainless-steel autoclave and heated to 180 °C for 24 h. After cooling to room temperature, the precipitate was washed four times using deionized water via centrifugation, then dried at 50 °C for 12 h. The M–$MoS_2$ (M = Pt, Pd, Au) catalysts were synthesized as follows. In brief, 40 mg of $MoS_2$ powder was mixed with 30 ml $H_2O$ in a round-bottom flask, and the mixture was ultrasonicated for 1 h; then each precursor solution (Pd (OAc)$_2$, $H_2PtCl_6$, $H_2AuCl_4$, Alfa Aesar) was added respectively and heated to 60 °C for 12 h. The products were obtained by filtration of the suspension, followed by dialysis in deionized water.

**Materials characterization**. The HAADF-STEM images were obtained by using a Titan 80–300 scanning/transmission electron microscope operated at 300 kV,

equipped with a probe spherical aberration corrector. In-situ STEM-EELS was performed using a JEOL 3000 F TEM without Cs corrected. TEM, HAADF-STEM and EDX mapping were tested on A Philips TECNAI G2 electron microscope operating at 200 kV accelerating voltage. SEM images were taken using a XL 30 ESEM FEG field emission scanning electron microscope. Mo and Pd K-edge X-ray absorption spectra were performed at the BL14W1 beamline of the Shanghai Synchrotron Radiation Facility, operating at 3.5 GeV with injection currents of 140–210 mA.[41] Si (111) and Si (311) double-crystal monochromators were used to reduce the harmonic component of the monochrome beam. Mo and Pd foils were also tested in transmission mode as references. The Mo L$_3$-edge XANES spectra were tested at the 4B7A beamline of the Beijing Synchrotron Radiation Facility (BSRF), China, in total electron yield (TEY) mode, where the sample drain current was collected under pressure smaller than $5 \times 10^{-8}$ Pa. The beam from a bending magnet was monochromatic with a varied line-spacing plane grating and was refocused by a toroidal mirror. A Bruker ER 200D spectrometer was used to test the ESR, and the measurements were performed at room temperature. An AXIS Ultra DLD (Kratos Company) was used for XPS measurements, using a mono-chromic Al X-ray source. PALS data were collected on a fast−slow coincidence ORTEC system, and the time resolution was approximately 195 ps (full width at half-maximum). XRD measurements were performed on a PW-1700 diffractometer using a Cu Kα ($\lambda = 1.5405$ Å) radiation source (Philips Co.). Raman spectra were collected on a J-Y T64000 Raman spectrometer with 514.5 nm wavelength incident laser light. Elemental analyses were collected by ICP-AES-MS (Inductively Coupled Plasma-Atomic Emission Spectroscopy-Mass Spectrometry) using a Thermo Elemental IRIS Intrepid.

**Electrochemical measurements**. The electrochemical performance was measured in a $N_2$-saturated $H_2SO_4$ solution (0.5 M) using a standard three-electrode setup using Princeton Applied Research. The glassy carbon electrode (3 mm in diameter) coated with the catalysts served as the working electrode, a saturated calomel electrode (SCE) served as the reference electrode, and a graphite plate served as the counter electrode. Inks were prepared by ultrasonically dispersing 4 mg of the catalysts ($MoS_2$, Pd–$MoS_2$, Pd–C, and Pt–C) in a suspension containing 20 μL of a Nafion (5wt%) solution and 1000 μL ($V_{ethanol}/V_{Ultrapure water} = 10:9$). The catalyst loading was calculated as approximately 0.222 mg cm$^{-2}$, where the geometric area of the glassy carbon electrode used was 0.07065 cm$^2$. To calculate the TOF of the catalyst, we also optimized the loading of the catalyst. The HER performances were tested in $H_2$-saturated 0.5 M $H_2SO_4$ using the linear sweep voltammetry at a scan rate of 2 mV s$^{-1}$. All data presented were iR corrected, where the solution resistances were determined by EIS experiments. The potential values shown were with respect to the reversible hydrogen electrode (RHE).

**DFT calculations**. All first-principles calculations were implemented within the framework of DFT in the Vienna ab initio Simulation Packageusing (VASP).[42–45] The exchange-correlation interactions were treated within the generalized gradient approximation of the Perdew-Burke-Ernzerhof (PBE) type.[46–48] The plane-wave cutoff energy was 400 eV and a k-mesh of $3 \times 3 \times 1$ was adopted to sample the Brillouin zone. Lattice geometries and atomic positions were fully relaxed until the forces on each atom were <0.01 eV/Å and the convergence threshold for energy was $10^{-4}$ eV. Vacuum layers of 15 Å were introduced to minimize interactions between adjacent layers in all supercells.

**Data availability**. The data that support the findings of this study are available from the authors on reasonable request; see author contributions for specific data sets.

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

## Acknowledgements

The work is supported by the National Natural Science Foundation of China (21633008), the National Key R&D Program of China (Grant No. 2017YFA0204800) the Strategic Priority Research Program of CAS (XDA09030104), Jilin Province Science and Technology Development Program (20160622037JC), Natural Science Funds of China (21525311 and 21773027). We thank the computational resources at the SEU and National Supercomputing Center in Tianjin. J.G. thanks for the Hundred Talents Program of Chinese Academy of Science.

## Author contributions

W.X., J.G., J.W., and Z.J. co-supervised the whole work. Z.L. and M.X. contributed to the synthesis of material and the characterization. Z.L., M.X., W.X., J.G. and C.L. contributed to analysis of the electrochemical experiments results. Y.O. and J.W. contributed to the theory calculation. H.Z and Z.J. contributed to the X-ray absorption fine structure spectroscopy and total electron yield spectroscopy. D.T. contributed to the In-situ STEM - EELS,. X.C. contributed to the PALS. The manuscript was primarily written by Z.L., W.X., J.G., J.W and Z.J. All authors contributed to discussions and manuscript review.

## Additional information

**Competing interests:** The authors declare no competing interests.

