## [Peer Review File · Nature Communications]

Reviewers' comments:

Reviewer #1 (Remarks to the Author):

In this manuscript, the authors report a new method for substitutionally doping the basal plane of MoS₂ with Pd, which improves its activity as an electrocatalyst for the hydrogen evolution reaction. Their synthesis technique relies on a spontaneous MoS₂/Pd (II) redox reaction, which results in Pd replacing some Mo in the MoS₂ lattice. The authors report that this material has high HER activity, which they believe is due to its large active site density, high activity of each active site, and high electronic conductivity.

Overall, I believe this work represents an interesting new strategy for engineering electrocatalyst activity. The novelty of this approach of substitutionally doping the MoS₂ with Pd and the very high HER activity achieved by this technique will likely make this work of interest to many readers. However, I believe that in its present form, this study has several technical shortcomings. I am not convinced that all of the authors' claims are supported by the data they present in this manuscript. I have provided a detailed description of my specific concerns below. If the authors are able to revise this paper to address these concerns, I believe it may become appropriate for publication in Nature Communications.

Specific concerns:

1. I find the phenomenon of Pd doping into MoS₂ to be surprising. The authors have done extensive characterization to show how this works, but I still have a few remaining questions about the process. In the Supplementary Information page 5 section 3, the authors write "Furthermore, the oxidation of all Mo (III) into Mo (IV) and the leaching of Mo ions (due to law of conservation of charge) together unambiguously demonstrate the reducing capability of Mo (III)." First, the meaning of this sentence is not clear – what is the reducing capability of Mo (III)? Second, I do not see that the authors presented any experimental evidence of the Mo leaching into solution. This type of evidence would help to strengthen their argument about the mechanism. Can the authors measure Mo in solution after this process using a technique such as ICP-OES?

2. The authors' interpretation of the TEM images in Figure 2E and Supplementary Figure S8 are not completely convincing. In Figure S8, I can see no scale bar in this image which makes it impossible to assess the validity of the conclusions that the authors have drawn. In addition, it looks like the aspect ratio might be off – is this image stretched horizontally? In Figure S8, there are many regions of different contrast – some are uniformly brighter, some are uniformly darker, some are in between; in some places a regular hexagonal lattice is observed, in some places it is not observed. There are certainly some regions where a perfect crystalline atomic lattice is not visible, so I am skeptical of the statement "However, Pd species are excluded from occupying the S sites, as no distinct contrast alternation in hexagon or trigonal pattern is observed after Pd doping," from page 8 of the main text. I am also skeptical of the 1T/2H phase assignment in Figure 2E. Can the authors account for all the sources of contrast in Figure S8 based on their hypotheses about the nature of this material? I am concerned that the present analysis may appear as though the authors are "cherry picking" a couple of very small regions of their data that happen to appear to show these phenomena. Also it is not clear where within Figure S8 the Figure 2E image was taken from.

3. What prevents the incorporation of additional Pd into the film, and how is this controlled? To phrase this question differently, what limits the extent of the Pd incorporation reaction? Is it the initial concentration of Mo (III) species? If this is the case, can the authors change the original MoS₂ synthesis to modulate the Mo (III) concentration within the material, and show that this correlates

with additional Pd incorporation?

4. On PDF page 9, the authors say that the electron spin resonance data suggest that the 1%Pd-MoS₂ has approximately 3 times as many unsaturated sites with unpaired electrons as the pristine MoS₂, which corroborates the "formation of abundant SVs." Previously on page 7 the authors stated that "The Pd introduction does not induce observable morphological changes (Supplementary Fig. 2-3) to the MoS₂ (5-8 layers) nanosheets." Is the presence of these increased SVs observable with high resolution aberration-corrected TEM?

5. Overall, the authors' explanation of the reaction mechanism for the Pd substitutional doping into the MoS₂ is somewhat confusing. I suggest that it might be helpful to begin this discussion by including an overall reaction for the whole process, and discussing the thermodynamics for this overall reaction.

6. How were the electronic conductivity data shown in Figure 2h collected? I can't find an explanation for the sample preparation method, morphology, or measurement technique.

7. For the HER activity characterization, the onset potential should be defined based on a specific current density (for example, 0.1 mA/cm² or 0.5 mA/cm²) to be meaningful.

8. The experimental data do not justify the authors' assertions that the high HER activity of this material arises from its high electrical conductivity, high active site density, and high turn over frequency per active site. The contribution of each of these factors is not individually quantified in the experimental measurements. I believe that at least, the authors need to estimate the turn over frequency per active site from their experimental data to support their assertion that the intrinsic activity of each site has been improved. It is possible that just the overall site density has increased substantially, resulting in the better geometrical area-normalized activity. While the DFT calculations suggest that it is likely that the S atoms near the Pd dopants in the MoS₂ have a high turn over frequency, this needs to be confirmed experimentally. The authors should use their extensive structure characterization to estimate the active site density and determine the turn over frequency of each active site.

Reviewer #2 (Remarks to the Author):

This work presents a method of doping Pd atoms into MoS₂ basal plan for improved hydrogen evolution reaction activity. I cannot accept this paper for publication in Nature Communications due to the following problems in this work.

1. Impact. The author mentioned about the community's efforts in replacing noble metal catalysts such as Pt for large-scale deployment in practice. However, what is the meaning or significance of this work to replace Pt using noble Pd, with similar price and scarcity but much lower HER performance compared to Pt. There are many papers published before using a small amount of Pt loading (1%), but showed a much higher activity than this current work [see Nat. Commun. 7, 13638 (2016)]. In addition, the stability of Pd-MoS₂ does not meet current standards in the community for a high-impact journal. In Fig. S12, with a starting overpotential of 89mV to maintain a small scale 10 mA/cm² current, only in 10 h the overpotential was increased to ~ 120mV. I have to say that the 120 mV overpotential was supposed to deliver a current of ~ 20 mA/cm² as shown in Figure 3a, with a nearly 50 % of the current decreased. Hundreds of hours and hundreds of milliamps of stability tests have been shown in water-splitting, and the stability performance in this work is far below the criteria of a promising catalyst candidate.

2. Evidence. The authors claim the substitution of Mo atoms by Pd in solution with the followed

assumption of Pd coordination as active sites. However, I could not find a convincing, or direct evidence to support the major claim in this paper. No high-quality TEM/STEM or other characterizations and analysis were performed to demonstrate the position of Pd atoms on MoS₂. There could be many possible reactions happened during the deposition of Pd onto MoS₂ flakes, and I am skeptical about their proposed mechanism. First of all, they use thermodynamic redox potential to support that the substitution reaction could happen. However, this process could have an extremely high kinetic barrier to overcome. In fact, MoS₂ is a very stable compound existing in nature. Pd ions need to cleave three strong Mo-S covalent bonding to finish the substitution process under only mild conditions; however they could be easily doped into S vacancies existing in MoS₂ flakes, or aggregate into ultrasmall clusters on some defective sites or edge sites of the MoS₂ nanoflakes. The reason I believe there are S vacancies in their MoS₂ is the Mo to S ratio of 1:1.87 they have reported.

Reviewer #3 (Remarks to the Author):

This manuscript reports low overpotentials for the hydrogen evolution reaction on Pd-doped MoS₂. The activities and stabilities reported are impressive and appear to be some of the highest for Pt-free preparations. I have a number of concerns with the manuscript that should be addressed, but overall I found this an interesting work.

- The introduction states that they "rationally design" this catalyst, but the rest of the manuscript sounds more like it was just a lucky try (which is fine). E.g., if the catalyst were rationally designed I would expect the DFT calculations of binding energy to lead the discussion, and then the discussion would follow about how to design such active sites in practice. In the current work, they dive into very detailed synthesis and characterization details and the binding energy / mechanistic discussions are somewhat of an afterthought. Perhaps I am not interpreting what took place correctly, but if I were the authors I would tone down the "rational design" language or otherwise arrange the manuscript appropriately.

- The use of 'onset potential' is confusing, and rather arbitrary depending on who is reading the polarization curves. Most overpotentials are reported at a common metric such as -10 mA/cm², like in the work of Jaramillo and co-authors (DOI: 10.1021/cs500923c), in which a systematic comparison of various MoS₂ preparations was made. It does not appear to me that the onset potential for 1% Pd-MoS₂ is 0 V vs. RHE, which is what the authors report. I would suggest reporting all overpotentials at -10 mA/cm² as well as exchange current densities, but excluding the onset potentials. I would also recommend reporting per-site activities as was done in the Jaramillo paper above.

- The authors frequently claim that they have the highest activity "among heteroatom-doped MoS₂", but conspicuously leave out comparisons to non-doped MoS₂ preparations. I think the relevant comparison should be to the broader field of MoS₂ catalysis, not just doped preparations.

- Perhaps I didn't understand, but it seems that the authors don't have a consistent message on what they think the active site is in their catalyst. At many points the sulfur vacancies are discussed as catalytically active sites, but in other places (notably the DFT results) S bound to Pd is identified as the active site. It is fine to be uncertain about the active site, but this uncertainty should be more clearly conveyed to the reader.

- Figure S10a shows polarization curves for Pd-doped MoS₂ at varying concentrations. For this series, the HER performance increases with increasing Pd concentration. However, figure 3a shows that pure Pd/C is an inferior HER catalyst compared to Pd-doped MoS₂. At what concentration of Pd would you expect to see a decrease in HER performance? Is this decrease in HER performance attributed to the

formation of 'Pd islands' on MoS₂ rather than substitution of Mo atoms in MoS₂? It would be beneficial to elaborate on this result.

- Incorrect math in section 2 (subsection 1; redox reaction Mo and Pd) of the supporting information: the standard reduction potential of Mo⁴⁺/Mo³⁺ is multiplied by the stoichiometric coefficient, but E⁰ is an intensive property. However, this does not change the result (i.e. it's still a spontaneous process).

- Figure 3e (catalyst stability) is not iR-corrected. The caption does not mention iR-correction for the polarization curves in figure 3a; are these results iR-corrected?

- Are hydrogen binding energies in DFT calculated using the computational hydrogen electrode (CHE)? I assume so, but the method is neither mentioned nor cited.

- Figure 4d (HBE on different systems): The y-label says free energy, but not at which potential. The overall reaction ($H^+ + e^- \rightarrow 1/2 H_2$) is only thermoneutral at 0.0 V vs. SHE/RHE.

- In Figure 3 the curves are only identified by color, and the colors change from panel to panel. This makes it difficult to quickly read, and perhaps impossible for colorblind readers.

- A minor point, but a couple of references are made to "laws" of defect chemistry. Maybe a word like "principles" would be more appropriate?

- Overall, the manuscript has a number of language issues and could use a solid edit.

Finally, I should note that much of the manuscript is devoted to a discussion of characterization techniques that I have not employed myself; I hope that other reviewers critically assess this portion of the manuscript.

Reply to Reviewer 1 and revisions made accordingly (Italic words are the comments from the reviewer):

“In this manuscript, the authors report a new method for substitutionally doping the basal plane of MoS₂ with Pd, which improves its activity as an electrocatalyst for the hydrogen evolution reaction. Their synthesis technique relies on a spontaneous MoS₂/Pd (II) redox reaction, which results in Pd replacing some Mo in the MoS₂ lattice. The authors report that this material has high HER activity, which they believe is due to its large active site density, high activity of each active site, and high electronic conductivity.

t if I were the authors I would tone down the "rational design" language or otherwise if I were the authors I would tone down the "rational design" language or otherwise arrange the manuscript appropriately.”

MoS₂ with Pd and the very high HER activity achieved by this technique will likely make this work of interest to many readers. However, I believe that in its present form, this study has several technical shortcomings. I am not convinced that all of the authors' claims are supported by the data they present in this manuscript. I have provided a detailed description of my specific concerns below. If the authors are able to revise this paper to address these concerns, I believe it may become appropriate for publication in Nature Communications.”

Reply: We thank the reviewer for the positive remarks and for many constructive suggestions to help us further improve our manuscript.

“1. I find the phenomenon of Pd doping into MoS₂ to be surprising. The authors have done extensive characterization to show how this works, but I still have a few remaining questions about the process. In the Supplementary Information page 5 section 3, the authors write “Furthermore, the oxidation of all Mo (III) into Mo (IV) and the leaching of Mo ions (due to law of conservation of charge) together unambiguously demonstrate the reducing capability of Mo (III).” First, the meaning of this sentence is not clear – what is the reducing capability of Mo (III)? Second, I do not see that the authors presented any experimental evidence of the Mo leaching into solution. This type of evidence would help to strengthen their argument about the mechanism. Can the authors measure Mo in solution after this process using a technique such as ICP-OES?”

Reply:

Regarding the first question:

We apologize for puzzling the reviewer due to our inappropriate description. By saying “the reducing capability of Mo(III)”, we actually mean that Mo in valence (III) can act as the reducing agent towards the reduction of Pt, Au, and Pd, etc. The

reducing power can be demonstrated both theoretically and experimentally. First, according to Equation 1, the standard redox potential (SRP) between Mo(III) and Mo(IV) is located at -0.04V, which can theoretically reduce any redox couples with higher SRP (Pd/Pd²⁺, Au/Au³⁺, Pt/Pt⁴⁺, etc). Second, we carried out a series of experiments to validate the redox power of Mo (III). We mixed the pristine MoS₂ sample with Pt(IV) and Au(III) complex solutions for reaction, respectively. As shown in **Figure R1**, we found that in comparison with a mixed Mo(III) and Mo(IV) valence states in the pristine sample, the final samples of Pt-MoS₂ and Au-MoS₂ only show Mo in IV valence state, indicating the complete oxidation of all Mo(III) into Mo(IV). In the meantime, the Pt(IV) and Au(III) precursors were all converted into Pt(0) and Au(0) metallic states in the resultant Pt-MoS₂ and Au-MoS₂ hybrids. Thus, it is confirmed that the reactions occurred through Equation 2 and 3 in the following, where Mo(III) serves as a reducing agent to reduce a class of transition metal ions (such as Pt(IV), Au(III), Pd(II)). The relative content has also been corrected in the supplementary Information.(**Page 5-6**)

Figure R1. (a) High-resolution XPS results (Mo 3d region) of Au-MoS₂, Pt-MoS₂, MoS₂ and Pd-MoS₂. (b) High-resolution XPS results (Au 3d region) of Au-MoS₂. (c) High-resolution XPS results (Pt 3d region) of Pt-MoS₂.

Regarding the second question:

We fully agree with the reviewer that the evidence of Mo leaching into the solution is very much useful. According to the reviewer's suggestion, we

resynthesized the Pd-MoS₂, Pt-MoS₂ and Au-MoS₂ samples, where the reaction condition was strictly controlled, as illustrated in the Methods shown in the manuscript (Page 17). For comparison, the pristine MoS₂ was also treated without the introduction of Pd in the same condition. The mixed solutions were then subjected to filtration, where the final solution was collected and tested through ICP-OES (see Table R1). For the pristine MoS₂ sample, no Mo or S ion was detected in the solution, suggesting the high anti-dissolution capability of MoS₂. On the contrary, after the reaction between Pd and MoS₂, we found that Pd was completely consumed, while both Mo and S ions were detected in the solution. The presence of both ions in the solution can be easily explained through Equation 4-6, as well as Figure R1. As is clear from Equation 4 that the oxidation of Mo(III) by Pd (II) will result in leaching of Mo ion from the MoS₂ bulk due to the principles of conservation of charge, the re-reduction of Mo(IV) due to the spontaneous reaction shown in Equation 6 results in the formation of S vacancies. We do not rule out the possibility that the leached ions of Mo and S may form clusters that dispersed in the solution, which however does not contradict our claim on the happening of an interfacial reaction. Further, the XPS results of Pd-MoS₂ also provide strong evidence of the interfacial reaction, where the Mo atoms exhibit in a mixed valence states of III and IV, and Pd atoms are all presented in II rather than in metallic states. For Pt and Au, after redox reaction, the Mo ions were also detected by ICP-OES. Nevertheless, as no further reaction occurs after their reduction into metallic states, no leaching of S was observed. These experiment results are now added to the supporting information (Supporting information, Table S3)

$$E^{\theta} = -0.04\text{V} - (-0.754\text{V}) = 0.75\text{V}$$

$$\Delta rG^{\theta} = -nE^{\theta}F = -2 \times 0.75 \times 96500\text{J} = -144.75\text{kJ} < 0$$

Table R1. ICP-OES results of dissolved S, Mo, Pt, Au and Pd ions in solution after this redox process

Dissolved ions into the solution ($\mu\text{g/mL}$)	Mo	S	Pd	Au	Pt
Pd-MoS ₂	70.02	41.18	0	-	-
MoS ₂	0	0	-	-	-

Pt-MoS ₂	82.17	0	-	-	0
Au-MoS ₂	77.63	0	-	0	-

“2. The authors’ interpretation of the TEM images in Figure 2E and Supplementary Figure S8 are not completely convincing. In Figure S8, I can see no scale bar in this image which makes it impossible to assess the validity of the conclusions that the authors have drawn. In addition, it looks like the aspect ratio might be off – is this image stretched horizontally? In Figure S8, there are many regions of different contrast – some are uniformly brighter, some are uniformly darker, some are in between; in some places a regular hexagonal lattice is observed, in some places it is not observed. There are certainly some regions where a perfect crystalline atomic lattice is not visible, so I am skeptical of the statement “However, Pd species are excluded from occupying the S sites, as no distinct contrast alternation in hexagon or trigonal pattern is observed after Pd doping,” from page 8 of the main text. I am also skeptical of the 1T/2H phase assignment in Figure 2E. Can the authors account for all the sources of contrast in Figure S8 based on their hypotheses about the nature of this material? I am concerned that the present analysis may appear as though the authors are “cherry picking” a couple of very small regions of their data that happen to appear to show these phenomena. Also it is not clear where within Figure S8 the Figure 2E image was taken from.”

Reply: Regarding the aspect ratio and the scale bar of Figure S8, we have re-processed the image of Figure S8 according to the suggestion (Figure R2c below).

Regarding the regions of different contrast in the TEM images:

Here we need to first explain the material properties of our homemade MoS₂. Since our MoS₂ was chemically synthesized through a hydrothermal technique, the final sample shows a flower-like two-dimensional nanosheet structure (Figure R2a) with both monolayer and multilayer flakes being observed through high-resolution TEM (HRTEM) and atomic force microscopy (AFM). Therefore, unlike the well-defined single layer structure reported in literature (Manish Chhowalla, et al reported in Nat Nanotechnology, 10, 313; Nørskov, et al stated in Nature Materials, 15, 364; Zhang Hua, et al reported in Nature Chemistry, 5, 263), the multi-dispersion nature of our catalyst makes it extremely hard to obtain high quality images. However, according to the suggestions from the reviewer, we have retested the sample and tried our best to obtain images with higher quality, and the results are now shown in Figure R2 and Figure R3. The way we choose the testing area is shown from Figure R2a to d, where the edge areas were firstly selected to get HRTEM images with thinner layers.

As shown in **Figure R2c and d**, the brighter regions can be ascribed to the presence of multilayer MoS₂, which has also been observed previously by Tsang reported in Nature Chemistry, 2017, 9, 810 (**the image is also shown in the following**). The uniformly darker regions can be found when the electron beam exposure time was over 60 s. In order to minimize such knock-on damage, the electron beam exposure time was limited to 20 s for each region of the specimen investigated (Nature materials, 15, 364). Furthermore, we have to mention that the Pd redox doping can induce more Mo vacancies and S vacancies, which can result in regions where a perfect crystalline atomic lattice is not visible.

Regarding “the displacement of the Mo by Pd” and “the coexistence of the 1T and 2H phases”:

For the question “the displacement of the Mo by Pd”, we have removed the inappropriate description. The exact configuration of Pd doping in the image is actually confirmed by a number of characterization techniques, among which the extended X-ray absorption fine structure (EXAFS) provides the most direct evidence. From the Fourier transform of k³-weighted Pd K-edge EXAFS spectra, the disappearance of first-shell Pd-Pd scattering peak at 2.51 Å in comparison with Pd foils indicates that Pd species are not in the form of metallic Pd nanoparticles or clusters. A prominent peak centered at much lower R position is observed at 1.84 Å instead, corroborating the dominance of Pd-S scattering contribution. The fitting results of the Fourier transform plot exhibits a Pd-S bond distance of 2.33 Å and a Pd to S coordination number of 4.33. According to Fig. 2c and Fig. 2d, the absence of first shell Pd-Mo scattering excludes the possibility of Pd to direct bonding with Mo, thereby confirming that Pd is not occupying S sites in MoS₂. Thus, it is confirmed that Pd occupies the Mo sites. Therefore, although the HRTEM does not provide direct evidence on the doping position of Pd, it does not affect our claim that Pd atoms actually replace the Mo sites.

As for the coexistence of 1T and 2H phase, we have to claim that the selection of areas is actually accomplished by the flow shown in **Figure R2**. Due to the multilayer nature of the MoS₂, we have to choose the relative thinner areas to distinguish the phases. However, the coexistence of the 1T and 2H area is universally found rather than deliberately “cherry picking” the relevant area. In order to show such characteristic of the Pd-MoS₂, we reexamined the 1%Pd-MoS₂ by the sub-angstrom resolution aberration-corrected HAADF-STEM microscopy, with results shown in following images in **Figure R3**. In larger areas, the coexistence of 1T and 2H phases is more clearly evidenced. Furthermore, Raman spectra provides another solid evidence for the coexistence of 1T and 2H phase, where characteristic peaks corresponding to each phase are clearly noticed.

TEM corresponding mark. (c). Sub-angstrom resolution aberration-corrected HAADF-STEM images of Pd-MoS₂ corresponding circle in (b). (d). HAADF-STEM image showing a layer of MoS₂ marked by tetragonal in (c).

Figure R3a&b Sub-angstrom resolution aberration-corrected HAADF-STEM images of 1%Pd-MoS₂.

Tsang in Nature Chemistry, 2017, 9, 810-816:AC-TEM Characterization

Supplementary Fig. 5 (a, b) Termination of a MoS₂ layer. HAADF-STEM image showing a layer of MoS₂ oriented turbostratically to those underneath, showing the characteristic moiré pattern. **(c) MoS₂ bilayer.** 60 kV ADF-STEM image, showing a stacked bilayer region atop to a monolayer of MoS₂.

“3. What prevents the incorporation of additional Pd into the film, and how is this controlled? To phrase this question differently, what limits the extent of the Pd incorporation reaction? Is the initial concentration of Mo (III) species? If this is the case, can the authors change the original MoS₂ synthesis to modulate the Mo (III) concentration within the material, and show that this correlates with additional Pd incorporation?”

Reply:

This is a very good question. The presence of Mo (III) species in the MoS₂ matrix is indeed necessary for the introduction of Pd into the MoS₂ backbone, where we believe the Pd fixation is accomplished through the interfacial redox reaction. In the revised manuscript, in order to answer the reviewer’s question and further support our claim, we carried out two additional experiments independently.

i) We used the standard 2H MoS₂ sample (Mo: S=1: 2, without Mo (III) species) instead of the chemical synthesized MoS₂ (homemade pristine MoS₂ used in the manuscript, with stoichiometry at 1:1.87) to react with Pd (II). As showed in **Figure R4a-b** (shown in the following), the Mo L₃-edge XANES spectra and high-resolution XPS results (Mo 3d region) of 2H MoS₂ sample confirmed that all Mo atoms are present as Mo(IV). The reaction between standard 2H MoS₂ and Pd(OAc)₂ was controlled at the same condition as those illustrated in the Methods. The concentrations of Pd in the solution before and after the reaction were monitored through ICP-OES, as showed in **Table R2** presented in the following. Clearly, no obvious change in Pd concentration is observed. Meanwhile, the final 2H-MoS₂ sample after reaction with Pd was also tested through ICP-OES, where no Pd (under the detection limit) was obtained in the final sample. All these results confirm that standard 2H-MoS₂ is not capable of reducing Pd(II), further validating that Pd atoms are introduced through the redox mechanism. These experimental results are now added to the supporting information (**Supporting information, Page 6**).

ii) As for the question regarding the limits of Pd incorporation extent, we actually can increase the amount of Pd doping as long as there are sufficient surface Mo sites, due the reaction mechanism presented in Equation 4-6 shown above. Clearly, the incorporation of Pd occurs in a two-step mechanism, where after the reaction, Mo sites are reduced back into Mo (III) and resulting in the release of its reducing power. Therefore, theoretically, the content of Pd can be continually increased until the accessible Mo atoms are completely substituted by Pd atoms. However, it is worthy to note that the correlation between Pd content and the performance of the MoS₂ is not always positive, as shown in **Figure R5** and also presented below. While increasing Pd from 2% to 10% results in increased activity towards HER, further increase in Pd content causes performance decay. This can be explained as follows: in the Pd-MoS₂

catalyst, Pd atoms are not the active sites, and instead, they function by activating the MoS₂ basal plane through the introduction of sulfur vacancies (SVs) and by activating the adjacent S atoms (Pd-S*-Mo). However, a Pd content that is too high will result in a decrease in the site density of Pd-S*-Mo, and a S-vacancy concentration that is too high results in deviation of intrinsic activity from the optimal value (see *Nørskov, et al reported in Nature Materials,15,364; Linyou Cao, et al stated J Am Chem Soc,138,16632*), both of which hinder the final HER performance. These experiment results are now added to the supporting information (**Supporting information, Figure S13**).

Figure R4 a Mo L₃-edge XANES spectra of the home-made MoS₂ and standard 2H-MoS₂. b High-resolution XPS results (Mo 3d region) of the standard MoS₂.

Table R2. ICP-OES results of Pd in solution and samples in MoS₂ (2H) after reaction with Pd(II).

The amount of Pd in solution or sample ($\mu\text{g/mL}$ or wt%)	Pd
Solution before redox reaction	12.12
Solution after redox reaction	11.87
MoS ₂ (2H) after reaction with Pd(II)	Under detect limit

Figure R5 a Polarization curves of 2%Pd-MoS₂, 5%Pd-MoS₂, 10%Pd-MoS₂ and 15%Pd-MoS₂. b Current densities at overpotential of 0.075 V, 0.1 V and 0.125 V for Pd-MoS₂ with different Pd doping contents.

“4. On PDF page 9, the authors say that the electron spin resonance data suggest that the 1%Pd-MoS₂ has approximately 3 times as many unsaturated sites with unpaired electrons as the pristine MoS₂, which corroborates the “formation of abundant SVs.” Previously on page 7 the authors stated that “The Pd introduction does not induce observable morphological changes (Supplementary Fig. 2-3) to the MoS₂ (5-8 layers) nanosheets.” Is the presence of these increased SVs observable with high resolution aberration-corrected TEM?”

Reply: The high resolution aberration-corrected TEM can be used to observe the SVs only in well-defined MoS₂ monolayer samples. According to the reviewer’s suggestion, we have re-characterized the pristine MoS₂ sample and the Pd-MoS₂ sample using aberration-corrected HAADF-STEM microscopy. Unfortunately, no distinct difference can be observed from the images (**Figure R6**), indicating that the SVs cannot be successfully observed, probably due to the multilayer nature of the sample and the relatively low contrast of S in comparison to Mo.

Figure R6. (a) The sub-angstrom resolution aberration-corrected HAADF-STEM images of MoS₂. (b) The sub-angstrom resolution aberration-corrected HAADF-STEM images of 1%Pd- MoS₂.

“5. Overall, the authors’ explanation of the reaction mechanism for the Pd substitutional doping into the MoS₂ is somewhat confusing. I suggest that it might be helpful to begin this discussion by including an overall reaction for the whole process, and discussing the thermodynamics for this overall reaction.”

Reply: Thank the reviewer for his/her constructive suggestion. We have added the corresponding reaction mechanism for the Pd substitutional doping in the revised manuscript and for the convenience of the reviewers, we also copied it to this response as follows:

A two-step thermodynamically spontaneous reaction is expected. First, the redox process in Equation 3 is a thermodynamically spontaneous reaction ($E^\theta = 1.031V, \Delta G^\theta = -198.98kJ$), which leads to the reduction of Pd and oxidation of Mo, thereby creating Mo vacancies due to principles of conservation of charge. Afterwards, however, metallic Pd is thermodynamically favorable towards anchoring to the energetic Mo vacancies and spontaneously forming the more stable Pd-S bond ($K_{sp}=2.03 \times 10^{-58}$, Supplementary Table 2), as shown in Equation 4:) through its incorporation into the MoS₂ backbone ($E^\theta = 0.75V, \Delta G^\theta = -144.75kJ$). By injecting electrons into the MoS₂ substrates, Mo (IV) can be back reduced into Mo (III) and causes the leaching of S into the solution (principles of conservation of charge) and formation of S vacancies (see Equation 1-2).

“6. How were the electronic conductivity data shown in Figure 2h collected? I can't find an explanation for the sample preparation method, morphology, or measurement technique.”

Reply: We added detailed information about the conductivity measurement in supporting information as shown below:

The electrical resistances of samples were determined by using a homemade button cell (refer to Angew. Chem. Int. Ed. 2013, 52, 11755-11759). The powder samples were pressed in the mold with certain pressure and time (~10MPa, 5min). The sample is inserted between two smooth polished steel discs. Electrochemical impedance spectroscopy (EIS) performed at high frequency using Princeton Applied Research PARATAT MC. The operating frequency range was between 10 mHz and 10 kHz, the DC potential was 0 V compared to an open circuit, and the AC amplitude was 10 mV. In this case, the phase angle between the voltage applied and the current induced is zero; the impedance of the sample as a function of frequency is present as a horizontal line. The value of resistance of sample is equal to the impedance; and the resistance can be directly read from the $|Z|$ -axis in the Bode. The electronic conductivity was calculated according the formula:

$$\sigma = \frac{l}{RS}$$

Here, the l is the thickness of specimen, R is electrical resistance of specimen, S is the area of specimen. l and S can get from the mould. R is obtained from Bode spectra (Bode spectra is read from the electrochemical impedance spectroscopy).

“7. For the HER activity characterization, the onset potential should be defined based on a specific current density (for example, 0.1 mA/cm² or 0.5 mA/cm²) to be meaningful.”

Reply: In this work, we used the overpotential at 0.5 mA cm⁻² as onset potential.

“8. The experimental data do not justify the authors’ assertions that the high HER activity of this material arises from its high electrical conductivity, high active site density, and high turn over frequency per active site. The contribution of each of these factors is not individually quantified in the experimental measurements. I believe that at least, the authors need to estimate the turn over frequency per active site from their experimental data to support their assertion that the intrinsic activity of each site has been improved. It is possible that just the overall site density has increased substantially, resulting in the better geometrical area-normalized activity. While the DFT calculations suggest that it is likely that the S atoms near the Pd dopants in the MoS₂ have a high turn over frequency, this needs to be confirmed experimentally. The authors should use their extensive structure characterization to estimate the active site density and determine the turn over frequency of each active site.”

Reply: Thanks for the reviewer’s suggestion.

i) Regarding the individual contribution of the electrical conductivity, high active site density, and high turnover frequency per active site.

Due to the fact that the interfacial reaction between Pd and MoS₂ simultaneously causes the creation of S-vacancy, the conversion of 2H into 1T phase, and the creation of Pd bonded S sites, we are not able to completely separate and quantify the contribution of each factor by varying the experimental parameters in our catalysts. Nevertheless, a combined effect of the three factors on the final catalysis behavior can be confirmed by comparing the catalytic behavior of our catalysts with the reported values of the 2H-MoS₂ with S-vacancies, the 1T-MoS₂, and 1T-MoS₂ with S-vacancies reported in literature (see **Table R3**).

As far as we know, (1) the best activity of 2H-MoS₂ with S-vacancies is reported by Linyou Cao, et al in **J. Am. Chem. Soc. 2016, 138, 16632**, where the catalysts with optimal S-vacancies exhibits overpotential of 170 mV at 10 mA/cm². The catalyst shows maximum catalytic activity with SV densities varies between 7%~15%, and further increasing the SV concentration leads to performance decay. Xiaolin Zheng, et al also reported this view in **Nat Commun, 2017, 8, 15113** and **Nat. Mater,15,364**; (2) The lowest overpotential reported for 1T-MoS₂ at 10mA/cm² is ~200 mV (See **J. Am. Chem. Soc. 2016, 138, 7965-72**; **Nano Lett. 2013, 13, 6222-7**; **Nat. Commun. 2016, 7, 10672**; **J. Am. Chem. Soc. 2017, 135, 10274**); (3) Even in the case of 1T-MoS₂ with abundant S-vacancies and edges (the degree of defects exceeds 83.7%), the lowest overpotential ever reported is 153 mV(**Song Jin, et al reported in J. Am. Chem. Soc. 2016,138,7965**).

In our work, the 1%Pd-MoS₂/CP exhibit very low overpotential at 10 mA/cm² of only 78 mV, which is 75 mV smaller than the most active MoS₂ catalysts reported (1T-MoS₂ with abundant S-vacancies and edges, **J. Am. Chem. Soc. 2016,138,7965**). With the Tafel slope measured at 80 mV/dec, the decrease in overpotential

corresponds to approximately 10 times increase in active site density, provided that the sites only exhibit the same intrinsic activity with that of the S-vacancies and edges in 1T-MoS₂. This is impossible as the degree of the defects in the reported sample is already very high (83.7%) and the catalyst loading used for tests were similar with our tests (0.15 mg cm⁻² vs 0.22 mg cm⁻²). Therefore, besides the effect from the creation of S vacancies and the phase conversion from 2H to 1T, we believe that the creation of active site with higher intrinsic activity in our catalysts largely contribute to the overall HER performance. As the phase conversion is confirmed by Raman Spectra, the generation of more S vacancies is confirmed by ICP-OES and EPR, and the Pd-S* is confirmed as more effective active site by DFT, we have made a conclusion that the performance enhancement of the Pd-MoS₂ is a result of the combination of these three effects, rather than a single site density increment. The relative content has also been added in the supplementary Information. (see **Supporting Information subtitle “6. Generation of intrinsically more active S atop active sites (Pd bonded S)”**)

ii) Regarding the TOF per active site

Thanks to the reviewer for the constructive comment. We fully agree with the reviewer that the comparison of TOF per active site is the most accurate way to demonstrate the intrinsic activity of the catalysts. In the revised manuscript, we calculated the average TOF of our catalysts, and compared it with the reported values shown in the literature (Shown as **Table R4** and **Table R5**). The calculation results are now added to the supporting information as Table S10. The average TOFs of 1%Pd-MoS₂ are **0.15 s⁻¹, 2.77 s⁻¹, 7.84 s⁻¹, 9.10 s⁻¹, 16.54 s⁻¹ and 21.15 s⁻¹** at **0 V, 0.1 V, 0.15 V, 0.16 V, 0.2 V and 0.24 V**, respectively, which surpasses most of the state-of-the-art MoS₂ reported in **Table R5**. As far as we known, the highest TOF_{S-vacancy} ever reported for MoS₂ was **0.05~0.16 s⁻¹** at **0 V** (*Nat. Mater*, **15**, **364**), where a well-defined monolayer MoS₂ was supported on Au substrate, thus making all the SVs in the catalysts available at the reaction interface. For all other catalysts represented multicrystalline or amorphous structures, much inferior catalytic behaviors were observed (**Table R5**). For our 1%Pd-MoS₂ catalyst, it not only shows the highest TOF among all the multicrystalline or amorphous structured catalysts, but also represents TOF of **0.15 S⁻¹** at **0 V**, on the high end of the values reported in *Nat. Mater*, **15**, **364**. Taking into the account of the multicrystalline nature of our testing electrode, we can confirm that the much higher average TOF was obtained. Thus, the high average TOF of the 1%Pd-MoS₂ reflect that the increased catalytic activity is originated from both increase in active site density and the intrinsic activity of each site. We have made changes to the manuscript (see **page 15 “Discussion”**) and supporting information (see **Supporting Information subtitle “5. TOF calculation”** and **table S10**) accordingly.

The TOF was calculated according the following formula:

$$TOF = \frac{\text{Total number } H_2 \text{ atoms per second}}{\text{Total number of active sites per unit area}} = \frac{j/(2 \times q)}{N}$$

where $q = 1.6 \times 10^{-19}$ C is the elementary charge, and 2 accounts for 2 H atoms per H_2 molecule. The DFT calculations suggest that the S atoms near the Pd dopants in the MoS_2 are the prior adsorption sites, and the sulfur vacancies, although are inferior to the Pd-S* sites, are also catalytically active for HER. We use the total number of the sulfur vacancies and the Pd-S* sites in the MoS_2 as the number of active sites, which can be estimated from the number of Pd. The number of the sulfur vacancies is estimated using defect equation.

$$N_{S\text{-vacancy}} = N_{Pd}$$

The 1%Pd- MoS_2 loading (on a glassy carbon electrode) was $222 \mu\text{g} / \text{cm}^2$. The MoS_2 is dispersed on the electrode to form a film and we assume in the calculation that all Pd atoms are exposed on the surface. The atomic percentage of Pd was determined to be 1% from ICP-OES. Using the deposited mass, molar mass, and the atomic percentage of Pd atoms, the total numbers of Pd sites (N_{Pd}) and the total active sites ($N_{S\text{-vacancy} + Pd\text{-S}^*}$) are shown as follows:

$$N_{Pd} = \frac{222 \mu\text{g} / \text{cm}^2}{160.072 \text{g} / \text{mol}} \times 6.022 \times 10^{23} \text{mol}^{-1} \times 1\% = 8.35 \times 10^{15} \text{cm}^{-2}$$

$$N_{S\text{-vacancy} + Pd\text{-S}^*} = 1.67 \times 10^{16} \text{cm}^{-2}$$

TableR3. Comparison of HER activity for 1%Pd- MoS_2 with the 1T- MoS_2 , the 2H- MoS_2 with S-vacancies and 1T- MoS_2 with S-vacancies in acidic media.

Catalyst	Electrolytes	η (mV vs RHE) for $j = -10 \text{mAcm}^{-2}$	Current density (μAcm^{-2})	References
1% Pd-MoS_2	0.5 M H_2SO_4	89	805	This work
1%Pd-MoS_2/C P		78	-	
P-1T- MoS_2	0.5 M H_2SO_4	153	15.8	J. Am. Chem. Soc.

P-2H-MoS ₂		218	10.5	2016,138,7965
1T-MoS ₂		203	12.6	
2H-MoS ₂ with S-vacancies	0.5 M H ₂ SO ₄	320	-	Nat Commum,2017,8,15113
1T-MoS ₂ with edges	0.5 M H ₂ SO ₄	~200	-	Nanolett,2013,13,6222
1T-MoS ₂	0.5 M H ₂ SO ₄	175	-	Nat Commum,2016,7,10672
1T-MoS ₂ with SVs	0.5 M H ₂ SO ₄	200	-	Sci Rep,2016,6,31092
2H-MoS ₂ with SVs	0.5 M H ₂ SO ₄	170	40	J. Am. Chem. Soc. 2016,138,16632
1T-MoS ₂	0.5 M H ₂ SO ₄	~210	-	J. Am. Chem. Soc.135,10274

P-1T-MoS₂ refers to 1T-MoS₂ with abundant S-vacancies and edges (the degree of defects exceeds 83.7%); P-2H-MoS₂ refers to 2H-MoS₂ with abundant S-vacancies and edges.

Table R4 The intrinsic activity of each site in 1%Pd-MoS₂

	Total Current density (mA cm ⁻²)	The Current density contributed by Pd doping (mA cm ⁻²)	$\overline{\text{TOF}}$ (S ⁻¹)
0 V	0.805	0.774	0.15
0.1 V	14.756	14.756	2.77
0.15V	41.882	41.882	7.84
0.16V	48.622	48.622	9.10
0.2V	89.556	88.328	16.54
0.24V	114.33	112.96	21.15

Table R5 The TOF of other MoS₂-based catalysts reported in the literature.

Catalyst	TOF (S ⁻¹) at exchange current density	Overpotential(V)	Corresponding TOF(S ⁻¹)	Reference
MoS ₂	0.013	-	-	Nano lett.2013,13,1341
UHV-deposited MoS ₂ nanoparticles on Au(III)	-	0.1	1	Science,2007, 317,100
		0.16	10	
MoS ₂ nanoparticles	0.02	-	-	
Sub-monolayer [Mo ₃ S ₁₃] ²⁻ HOPG	-	0.2	3	Nat.Chem,2014 6,248-253
[Mo ₃ S ₄] ⁴⁺ cubanes/HOPG	0.07	0.3	1~5	J.Phys.Chem.C 112,17492-17498 (2008)
Amorphous MoS ₃ -CV film		0.22/0.24	0.8/2	Chem.Sci.2011,2, 1262
MoS ₂	-	0.3	3.83	J.Am.Chem.Soc.2 017,139,15479
Zn-MoS ₂	-	0.3	15.44	
Irradiated Au-MoS ₂	-	0.3	8.76	J.Am.Chem.Soc.2 015,137,7365
Double-gyroid MoS ₂	-	0.2	1	Nat.Mater.2012,1 1,963

MoO ₃ -MoS ₂ nanowires		0.272	4	Nano let.2011,11,4168.
Irradiated Au-MoS ₂	-	0.3	8.76	J. Am. Chem. Soc. 2015, 137, 7365
Amorphous MoS ₃	-	0.2	0.3	ACS Catal. 2012, 2, 1916
Defect-Rich MoS ₂ sheets	-	0.3	0.725	Adv.Mater.2013,2 5,5807
V-MoS ₂ /Au	0.05~0.16	-	-	Nat.Mater,15,364

Reply to Reviewer 2 and revisions made accordingly (Italic words are the comments from the reviewer):

“This work presents a method of doping Pd atoms into MoS₂ basal plan for improved hydrogen evolution reaction activity. I cannot accept this paper for publication in Nature Communications due to the following problems in this work.

Reply: We thank the reviewer for the reviewing our work and for the critical comments to further improve our manuscript. Below is our reply to the comments.

1. Impact. The author mentioned about the community’s efforts in replacing noble metal catalysts such as Pt for large-scale deployment in practice. However, what is the meaning or significance of this work to replace Pt using noble Pd, with similar price and scarcity but much lower HER performance compared to Pt. There are many papers published before using a small amount of Pt loading (1%), but showed a much higher activity than this current work [see Nat. Commun. 7, 13638 (2016)]. In addition, the stability of Pd-MoS₂ does not meet current standards in the community for a high-impact journal. In Fig. S12, with a starting overpotential of 89 mV to maintain a small scale 10 mA/cm² current, only in 10 h the overpotential was increased to ~ 120 mV. I have to say that the 120 mV overpotential was supposed to deliver a current of ~ 20 mA/cm² as shown in Figure 3a, with a nearly 50 % of the current decreased. Hundreds of hours and hundreds of milliamps of stability tests have been shown in water-splitting, and the stability performance in this work is far below the criteria of a promising catalyst candidate.”

Reply: By questioning the impact of the work, the catalytic activity of the Pd-MoS₂ in comparison with Pt based materials and its stability are the major concerns of the

reviewer. Here, we would like to the answer these two questions sequentially.

Regarding the catalytic activity:

Indeed, it is known that Pt is the most effective catalyst towards HER. And by reducing the catalyst loading for Pt, high activity can still be acquired as those shown in the reference paper suggested by the reviewer [see *Nat. Commun.* 7, 13638 (2016)]. Specifically, in the above mentioned paper, an atomic layer deposition method was adopted to synthesize Pt single atoms and nanoclusters supported on carbon, where the final Pt loadings were **2.1wt%** (ALD50Pt/NGNs) and **7.4wt%** (ALD100Pt/NGNs), respectively. The mass activities normalized to the Pt loading at overpotential of **0.05 V** were reported to be **10.1 A mg⁻¹_{Pt}** and **2.12 A mg⁻¹_{Pt}**, which is rather attractive. However, we have to mention that every coin has its two facets. While the mass activity of the catalysts can become very high for catalysts derived from ALD method, the price of the Pt precursor used is far more expensive than merely Pt, thus resulting in much higher catalyst cost and makes the technique not favorable for large-scale production. Furthermore, the cluster-based catalysts suffer a critical stability issue as these clusters are in high surface energy and are only immobilized by the support through electronic interaction, where no chemical bond is formed between the support and the catalysts. Therefore, it is safe to conclude that although enormous efforts have been paid and significant progress has been achieved in developing low Pt loading catalysts, till now, no method has ever been presented that simultaneously guarantee the price, synthesis simplicity, activity, and stability demands for a cost effective commercial Pt catalyst, thus leaving huge space for developing alternative catalysts.

In our work, we would like to emphasize that the novelty and impact of our contribution lie in two aspects. First, we aim to activate the MoS₂ catalysts through chemical doping, which is still current research hotspot (*Nat. Mater.*, **15**, 364). For the first time, we demonstrate that the basal plane of the MoS₂ can be activated by a thermodynamically spontaneous redox doping process due to the redox feature of Mo(III)/Mo(IV). The atomic scale mechanistic understanding for the Pd doping and the origin of the high intrinsic activity was discussed in-depth. Second, we have enormously raised the catalytic performance of MoS₂ based catalysts. The **78 mV** over potential at **10 mA cm⁻²** is indeed the highest value ever reported for the doped-MoS₂ catalysts, thus showing the effectiveness in our doping strategy. Actually, we are not the first to introduce precious metals into the MoS₂ catalysts, precious metals including Pt, Au, Ru, and Pd have all been tried in order to boost the catalytic activity of MoS₂. Varied degrees of enhancement have been achieved, and the values are compared in **Table R6**. The superiority of our catalysts is clear, where MoS₂ doped with only 1%wt Pd exhibits mass activity at **1.49A mg⁻¹_{Pd}**, **8.27A mg⁻¹_{Pd}** and **19.24 A g⁻¹_{Pd}** at over potential of **0.05 V**, **0.1 V** and **0.15 V**, the highest shown in the table and among heteroatom doped molybdenum sulfide. Furthermore, the mass activity also far exceeds that of the Pd/C catalysts, demonstrating the generation of

highly active catalytic sites other than metallic Pd. Though 1%-Pd-MoS₂ catalyst is not as active as the ALD-Pt/NGNs [*Nat. Commun.* 7, 13638 (2016)] catalysts based on single atoms and clusters, 1%-Pd-MoS₂ indeed demonstrate comparable performance with the ALD100Pt/NGNs catalysts, i.e., **1.49 A mg⁻¹_{Pd}** versus **2.12 A mg⁻¹_{Pt}**, which is already very promising. Further, we will demonstrate next that the 1%-Pd-MoS₂ actually possess higher stability than the ALD-Pt/NGNs catalysts.

Table R6 Comparison of HER mass activity normalized to the precious metal loading of our catalysts with other electrocatalysts in acidic media.

Catalyst	The loading of Pd(Pt) in catalyst (wt%)	Loading (mg/cm ²)	Electrolytes	Scan rate (mVs ⁻¹)	Potential (V)	Mass activity (A mg ⁻¹)	References
1%Pd-MoS ₂	1.0	0.0022	0.5M H ₂ SO ₄	2	0.05	1.49	In this work
					0.1	8.27	
					0.15	19.24	
1%Pd-C	1.0	0.0035	0.5M H ₂ SO ₄	2	0.1	0.914	
20%Pt-C	20	0.0714	0.5M H ₂ SO ₄	2	0.1	0.088	
Pt-MoS ₂	1.7	0.018	0.1M H ₂ SO ₄	5	0.14	0.55	Energy. Environ. Sci. 8,1594
Pt-MoS ₂	10.4	0.00728	0.5M H ₂ SO ₄	2	0.1	6.87	Nat Commun,2013, 4,1444
Pd-MoS ₂	24.5	0.01715			0.15	2.92	
Au-MoS ₂	18.3	0.01281			0.14	3.90	
Ru-MoS ₂	11.3	0.00791			0.16	6.32	
Pt-MoS ₂	36	0.027	0.5M H ₂ SO ₄	2	0.1	1.71	Nat Commun,2017,

							8,14548
--	--	--	--	--	--	--	---------

Regarding the stability:

In **Figure R7a**, a 10 h chronoamperometry test was carried out to demonstrate the stability of the 1%Pd-MoS₂. As commented by the reviewer, the starting over potential was 89 mV for the catalyst to reach 10 mA/cm². However, regarding the over potential at 10 h, we have to claim that the final potential point mentioned by the reviewer (~ 120 mV) is actually caused by the floating of the chronoamperometry curve due to the accumulation of the H₂ bubble on the electrode, which blocks the surface and acts like a performance decay. The same phenomenon was also observed previously in the literature (see *Adv. Mater*, 2013, 25, 5807-5813 and *Nat. Commun.* 7, 13638 (2016)). In order to better elucidate the catalytic stability, we further conducted a 100 h chronoamperometry test (see **Figure R7b**). Clearly, the 1%Pd-MoS₂ exhibits an outstanding long-term operational stability beyond 100 h with an observed potential increase of only **14 mV**. Moreover, we further examined the Pd-MoS₂ catalyst using XPS characterization after the above electrolysis test (**Figure R7d**). Neither the content nor the state of Pd was altered, suggesting that Pd is firmly integrated into the MoS₂ backbone and highly stable under electrolytic conditions. Meanwhile, as we have illustrated in the manuscript, the accelerated durability tests (ADTs) were also performed to assess the catalyst durability. As shown in **Figure R7c**(in the following), no obvious potential decay is observed after the ADT tests, which exceeds that of the Pt based catalysts, using ALD50Pt/NGNs for comparison. Specifically, it is noted that the Pd-MoS₂ exhibits only less than **1%** performance decay after **5000** cycles tests, the ALD50Pt/NGNs suffers **4%** loss after **1000** cycles. Thus, it is confirmed that the Pd-MoS₂ exhibits excellent stability. These experimental results are now added to the supporting information (**Supporting information, Figure S15 and Figure S16**).

Figure R7. (a) Chronoamperometry tests of 1%Pd-MoS₂ and MoS₂ at a current density of 10 mA cm⁻² (10 h). (b) Chronoamperometry tests of 1%Pd-MoS₂ at a current density of 10 mA cm⁻² (100 h). (c) Stability measurements for MoS₂ and 1%Pd-MoS₂ using accelerated degradation tests (5000 cycles). (d) High-resolution XPS results (Pd 3d region) of the Pd-MoS₂ before and after electrolysis at 10 mA/cm² for 100 h.

“2. Evidence. The authors claim the substitution of Mo atoms by Pd in solution with the followed assumption of Pd coordination as active sites. However, I could not find a convincing, or direct evidence to support the major claim in this paper. No high-quality TEM/STEM or other characterizations and analysis were performed to demonstrate the position of Pd atoms on MoS₂. There could be many possible reactions happened during the deposition of Pd onto MoS₂ flakes, and I am skeptical about their proposed mechanism. First of all, they use thermodynamic redox potential to support that the substitution reaction could happen. However, this process could have an extremely high kinetic barrier to overcome. In fact, MoS₂ is a very stable compound existing in nature. Pd ions need to cleave three strong Mo-S covalent bonding to finish the substitution process under only mild conditions; however they could be easily doped into S vacancies existing in MoS₂ flakes, or aggregate into ultrasmall clusters on some defective sites or edge sites of the MoS₂ nanoflakes. The reason I believe there are S vacancies in their MoS₂ is the Mo to S ratio of 1:1.87 they have reported.”

Reply:

Regarding the doping mechanism of Pd, we have actually done a lot of work in verifying the mechanism we have proposed. We agree with the reviewer that MoS₂ is a very stable compound existing in nature, however, the reactions between MoS₂ and metal ions including Pd(II), Au(III), and Pt(IV) are unambiguously observed. As clearly suggested from **Table R1**(in reply to the first reviewer) measured through

ICP-OES test, the leaching of Mo ions into the solution occurs in all three cases, well corresponding to what suggested from Equation ($4\text{Mo}^{3+} \rightarrow 3\text{Mo}^{4+} + 3e(\text{left}) + \text{V}_{\text{Mo}}^{///}$ (one Mo left)). For Pt and Au, as no further reaction occurs after their reduction into metallic states, no leaching of S was observed. In the case of Pd, however, further redox reaction corresponding to Equation ($2\text{Mo}^{4+} + \text{S}^{2-} + 2e \rightarrow 2\text{Mo}^{3+} + \text{V}_{\text{S}}^{//}$ (one S left)) leads to the leaching of S ions. In the manuscript, the thermodynamic redox potential was only used to predict that the substitution reaction can happen spontaneously. For the leaching of the ions into the solution, it is actually driven by law of conservation of charge in defect chemistry. As illustrated in above two equations, after the interfacial redox reaction between Mo and Pd, the MoS_2 matrix needs to maintain its electronic neutral nature and therefore needs to drive the excessive ions out of its matrix.

As for the quality of the TEM image, we need to explain the material property of our homemade MoS_2 at first. Since our MoS_2 was chemically synthesized through a hydrothermal technique, the final sample shows a flower-like two-dimensional nanosheet structure. Therefore, unlike the well-defined single layer structure reported in literature (*Manish, et al reported in Nature Nanotechnology, 10, 313-8; Nørskov, et al stated in Nature Materials, 15,364; Zhang Hua, et al reported in Nature Chemistry, 5,263-75*), the multi-dispersion nature of our catalyst makes it extremely hard to obtain high quality images. Also, it is noted that the Pd and Mo possess similar Z contrast due to their similar atomic weight; therefore, it is very hard, if not impossible, to distinguish Pd from Mo through the TEM image. Therefore, the exact configuration of Pd doping in the image is actually confirmed by a number of characterization techniques, including XPS, XRD, in-situ heat treatment TEM coupled with EELS, and extended X-ray absorption fine EXAFS. First, there are no nanoparticles or large clusters appeared in the TEM image (Supplementary Fig. 3), which rules out the possibility of forming Pd or PdS_x compounds on the surface. Further characterization from XRD and XPS results indicate the Pd was expectedly immobilized as Pd (II) in MoS_2 . Second, the results from in situ heat-treatment TEM coupled with electron energy loss spectroscopy (EELS, Supplementary Fig. 6) and X-ray Diffraction (XRD, Supplementary Fig. 7) tests show that Pd are mostly integrate into the MoS_2 backbone firmly, and not formed into PdS_x clusters on the MoS_2 nanoflakes. Finally and most importantly, extended X-ray absorption fine structure (EXAFS) evaluation was utilized to provide the most direct evidence for the doping position of Pd. From the Fourier transform of k^3 -weighted Pd K-edge EXAFS spectra, the disappearance of first-shell Pd-Pd scattering peak at 2.51 Å in comparison with Pd foils indicates that Pd species are not in the form of metallic Pd nanoparticles or clusters. A prominent peak centered at much lower R position is observed at 1.84 Å instead, corroborating the dominance of Pd-S scattering contribution. The fitting results of the Fourier transform plot exhibits a Pd-S bond distance of 2.33 Å and a Pd to S coordination number of 4.33. According to Fig. 2c and Fig. 2d, the absence of first shell Pd-Mo scattering excludes the possibility of Pd to directly bonding with Mo, thereby confirming that Pd is not occupying S sites in MoS_2 . Thus, it is believed that Pd atoms occupy the Mo sites. Therefore, although the HRTEM does not provide

direct evidence on the doping position of Pd, it does not affect our claim that Pd atoms actually replace the Mo sites.

Reply to Reviewer 3 and revisions made accordingly (Italic words are the comments from the reviewer):

“This manuscript reports low overpotentials for the hydrogen evolution reaction on Pd-doped MoS₂. The activities and stabilities reported are impressive and appear to be some of the highest for Pt-free preparations. I have a number of concerns with the manuscript that should be addressed, but overall I found this an interesting work.

Reply: We thank the reviewer for the positive remarks and many constructive suggestions to help us further improve our manuscript.

- The introduction states that they "rationally design" this catalyst, but the rest of the manuscript sounds more like it was just a lucky try (which is fine). E.g., if the catalyst were rationally designed I would expect the DFT calculations of binding energy to lead the discussion, and then the discussion would follow about how to design such active sites in practice. In the current work, they dive into very detailed synthesis and characterization details and the binding energy / mechanistic discussions are somewhat of an afterthought. Perhaps I am not interpreting what took place correctly, but if I were the authors I would tone down the "rational design" language or otherwise arrange the manuscript appropriately.”

Reply: According to the reviewer’s suggestion, we have toned down the "rational design" language in the revised manuscript (**Page 3**).

“- The use of 'onset potential' is confusing, and rather arbitrary depending on who is reading the polarization curves. Most overpotentials are reported at a common metric such as -10 mA/cm², like in the work of Jaramillo and co-authors (DOI: 10.1021/cs500923c), in which a systematic comparison of various MoS₂ preparations was made. It does not appear to me that the onset potential for 1% Pd-MoS₂ is 0 V vs. RHE, which is what the authors report. I would suggest reporting all overpotentials at -10 mA/cm² as well as exchange current densities, but excluding the onset potentials. I would also recommend reporting per-site activities as was done in the Jaramillo paper above.”

Reply: Thanks for the reviewer’s suggestion. According to the reviewer’s suggestion, we have excluded the onset potential in the revised manuscript.

Regarding the TOF, we fully agree with what the reviewer that calculating the per-site activities (turnover frequency) of MoS₂ is the most accurate way to demonstrate the intrinsic activity of the catalysts. In the revised manuscript, we calculated the average TOF of our catalysts, and compared it with the reported values shown in the literature

(shown as **Table R4** and **Table R5**). The calculation results are currently added to the manuscript (**Page 15 “discussion”**) and supporting information (**Page 8-9 subtitle “5. TOF calculation”** and **table S10** and **table S11**). Specifically, average TOFs of 1%Pd-MoS₂ are **0.15 s⁻¹**, **2.77s⁻¹**, **7.84 s⁻¹**, **9.10 s⁻¹**, **16.54 s⁻¹** and **21.15 s⁻¹** at **0 V**, **0.1 V**, **0.15 V**, **0.16 V**, **0.2 V** and **0.24 V**, respectively, which surpasses most state-of-the-art MoS₂ reported (**Table R5**). Detail information can be found in Reply to the first reviewer in this response letter (**Page 14-17**).

“- The authors frequently claim that they have the highest activity "among heteroatom-doped MoS₂", but conspicuously leave out comparisons to non-doped MoS₂ preparations. I think the relevant comparison should be to the broader field of MoS₂ catalysis, not just doped preparations.”

Reply:

According to the suggestion, we have compared the overpotential to reach 10 mA/cm² ($\eta@10 \text{ mA cm}^{-2}$) and exchange current density of our catalyst with those advanced MoS₂-based catalysts reported in literature. The 1%Pd-MoS₂ and 1%Pd-MoS₂/CP represent $\eta@10 \text{ mA cm}^{-2}$ at only 89 and 78 mV, respectively. As showed in the table, our catalysts exhibits lower overpotential at 10 mA cm⁻² than most MoS₂-based materials (**Table R6**). Although the overpotential is higher than reported MoS₂/CoSe₂ (Nat Commum, 2015, 6, 5982), we have to mention that it is well known that other transition metal chalcogens such as Ni, Fe, Co, etc exhibits higher HER activity than MoS₂, yet MoS₂ possesses much better anti-leaching capability than these compounds (**Nature Mater,15,197-203; Advanced Materials, 28, 92-7**). Moreover, our Pd-MoS₂ catalyst possesses much higher current density (approximately 11 times) than that of MoS₂/CoSe₂, and exhibits the highest performance among the MoS₂ based catalysts in acidic medium. The relevant contents and comparison are now included in the revised manuscript (see **Page 11**).

“- Perhaps I didn't understand, but it seems that the authors don't have a consistent message on what they think the active site is in their catalyst. At many points the sulfur vacancies are discussed as catalytically active sites, but in other places (notably the DFT results) S bound to Pd is identified as the active site. It is fine to be uncertain about the active site, but this uncertainty should be more clearly conveyed to the reader.”

Reply:

We apologized for our confusing message conveyed in this manuscript. Actually, there are two different catalyst sites in our catalyst. On the one hand, we have created more S vacancies during the synthesis of Pd-MoS₂, which is confirmed by the ICP-OES (change in MoS₂ stoichiometry from MoS_{1.87} to Pd_{0.01}MoS_{1.82}) and EPR characterizations. It is well known that SVs are the active sites in the MoS₂ catalysts

(**J. Am. Chem. Soc. 2016, 138, 16632; J. Am. Chem. Soc. 2016, 138, 7965-72; Nat. Mater, 15, 364**), and according to our DFT calculation, the ΔG_{H} of these SVs are 0.09 eV, suggesting their favorable HER catalytic behavior. Therefore, the increase in SVs concentration unambiguously contribute to the increased catalytic behavior, and we regard this as the increase in site density; On the other hand, according to the DFT calculation, the Pd adjacent S sites are also found favorable for hydrogen adsorption, with ΔG_{H} at as low as -0.02 eV acquired. Taking into account that HER performance has been enormously increased, and is far better than those reported for 1T-MoS₂ with abundant edge and vacancy sites (as illustrated in Table S11 shown above), we believe that the significant increase in the catalytic activity is due to a combination of increased SVs sites and the introduction of new Pd-S* sites in the 1T-phased material. In the revised manuscript, according the reviewer's suggestions, we have made this claim clarified. (see **Density functional theory calculations in the revised manuscript. Pages 14-15**)

“Figure S10a shows polarization curves for Pd-doped MoS₂ at varying concentrations. For this series, the HER performance increases with increasing Pd concentration. However, figure 3a shows that pure Pd/C is an inferior HER catalyst compared to Pd-doped MoS₂. At what concentration of Pd would you expect to see a decrease in HER performance? Is this decrease in HER performance attributed to the formation of 'Pd islands' on MoS₂ rather than substitution of Mo atoms in MoS₂? It would be beneficial to elaborate on this result.”

Reply:

According to the suggestion, we further increased Pd doping contents in MoS₂. It is worth noting that the correlation between Pd content and the performance of the MoS₂ is not always positive, as showed in **Figure R8a-b** presented in the following. While increasing Pd from 2% to 10% results in increased activity towards HER, further increase in Pd content results in performance decay. This can be explained as follows: in the Pd-MoS₂ catalyst, Pd atoms are not the active sites, and instead, they function by activating the MoS₂ basal plane through the introduction of sulfur vacancies (SVs) and by activating the adjacent S atoms (Pd-S*-Mo). However, a Pd content that is too high will result in a decrease in the site density of Pd-S*-Mo, and an S-vacancy concentration that is too high results in deviation of intrinsic activity from the optimal value, both of which hinder the final HER performance. These experimental results are now added to the supporting information (**Supporting information, Figure S13**)

In order to investigate whether or not the decrease in HER performance can be attributed to the formation of 'Pd islands' on MoS₂ rather than substitution of Mo atoms in MoS₂, the 15%Pd-MoS₂ catalyst was synthesized and examined by XRD and XPS. We also measured the XPS of 2%Pd-MoS₂ catalyst for comparison. As displayed in **Figure R8c-d**, neither 15%Pd-MoS₂ nor 2%Pd-MoS₂ shows the presence of Pd based crystalline phases, which rules out the possibility of forming Pd islands

on MoS₂.

Figure R8. (a) Polarization curves of 2%Pd-MoS₂, 5%Pd-MoS₂, 10%Pd-MoS₂ and 15%Pd-MoS₂. (b) Current densities at overpotential of 0.075 V, 0.1 V and 0.125 V for Pd-MoS₂ with different Pd doping contents.(c) XRD pattern of 15%Pd-MoS₂.(d) XPS patterns of 2%Pd-MoS₂ and 15%Pd-MoS₂.

“- Incorrect math in section 2 (subsection 1; redox reaction Mo and Pd) of the supporting information: the standard reduction potential of Mo⁴⁺/Mo³⁺ is multiplied by the stoichiometric coefficient, but E⁰ is an intensive property. However, this does not change the result (i.e. it’s still a spontaneous process).”

Reply: The related errors were corrected in the revised manuscript.

“- Figure 3e (catalyst stability) is not iR-corrected. The caption does not mention iR-correction for the polarization curves in figure 3a; are these results iR-corrected?”

Reply: Many thanks for the reviewer’s comments. All polarization curves in figure 3a have been iR-correction. We have added it in our revised manuscript.

“- Are hydrogen binding energies in DFT calculated using the computational hydrogen electrode (CHE)? I assume so, but the method is neither mentioned nor cited.”

Reply: Yes, the hydrogen binding energies in our DFT calculation are using the computational hydrogen electrode (CHE) model. We have added the computational

details in the Supplementary Methods:

“Additionally, the potential dependence of hydrogen adsorption is including using the computational hydrogen electrode model (Nørskov, J. K. et al. Origin of the overpotential for oxygen reduction at a fuel-cell cathode. *J. Phys. Chem. B* 108, 17886–17892 (2004); Peterson, A. A., Abild-Pedersen, F., Studt, F., Rossmeisl, J. & Nørskov, J. K. How copper catalyzes the electroreduction of carbon dioxide into hydrocarbon fuels. *Energy Environ. Sci.* 3, 1311 (2010).), where

$$\Delta G_{\text{H}^+(\text{aq})} + \Delta G_{\text{e}^-} = 1/2 \Delta G_{\text{H}(\text{aq})}$$

at a potential of $U=0\text{V}$ versus RHE.

“Figure 4d (HBE on different systems): The y-label says free energy, but not at which potential. The overall reaction ($\text{H}^+ + \text{e}^- \rightarrow 1/2 \text{H}_2$) is only thermoneutral at 0.0 V vs. SHE/RHE.”

Reply: We have added the corresponding potential it in our revised manuscript. For convenience, the figures are also shown here as follows:

Figure R9, Free energy versus the reaction coordinates of different active sites.

“- In Figure 3 the curves are only identified by color, and the colors change from panel to panel. This makes it difficult to quickly read, and perhaps impossible for colorblind readers.”

Reply: The relative content has been corrected in the revised manuscript, as follows.

“- A minor point, but a couple of references are made to "laws" of defect chemistry. Maybe a word like "principles" would be more appropriate?”

Reply: Thanks for the valuable comments from the reviewer and the related errors we could find were corrected in the revised manuscript.

“- Overall, the manuscript has a number of language issues and could use a solid edit.”

Reply: Thanks for the careful review. We have made a thorough check on the whole manuscript and carefully corrected the inappropriate expression accordingly. The language of this manuscript was polished by the help from Springer Nature editing service (Receipt code: Bopy-189-01230230830).

REVIEWERS' COMMENTS:

Reviewer #1 (Remarks to the Author):

The revisions the authors made to their manuscript successfully addressed my concerns, and I now believe this paper is suitable for publication in Nature Communications.

Reviewer #2 (Remarks to the Author):

The authors have now answered the questions and the paper is ready for publication

(Reviewer #3 provided a confidential comment, but the reviewer appears satisfied that the authors' thorough revisions have addressed the reviewer's concerns)

Reviewers' Comments:

Reviewer #1:

Remarks to the Author:

The revisions the authors made to their manuscript successfully addressed my concerns, and I now believe this paper is suitable for publication in Nature Communications.

Reviewer #2:

Remarks to the Author:

The authors have now answered the questions and the paper is ready for publication

Reviewer #3:

Remarks to the Author:

The authors provided a confidential comment, but the reviewer appears satisfied that the authors' thorough revisions have addressed the reviewer's concerns

Response:

We would like to thank all the referees for finding the revision satisfactory and suggesting to accept the paper for publication in Nature Communications. We again deeply appreciate all the referees for their constructive comments and suggestions in the first review, which are all very helpful in improving our manuscript.